# SiriuS: Self-improving Multi-agent Systems via Bootstrapped Reasoning

**Wanjia Zhao, Mert Yuksekgonul, Shirley Wu, James Zou**
Department of Computer Science
Stanford University

## Abstract

Multi-agent AI systems powered by large language models (LLMs) are increasingly applied to solve complex tasks. However, these systems often rely on fragile, manually designed prompts and heuristics, making optimization difficult. A key challenge in optimizing multi-agent systems is acquiring suitable training data for specialized agents. We introduce SIRIUS, a self-improving, reasoning-driven optimization framework for multi-agent systems. Central to our approach is the construction of an experience library: a repository of high-quality reasoning trajectories. The library is built by retaining reasoning steps that lead to successful outcomes, providing a robust training set for optimizing multi-agent system. Additionally, we introduce a library augmentation procedure that refines unsuccessful trajectories, further enriching the library. SIRIUS boosts performance by 2.86% to 21.88% on reasoning and biomedical QA and enhances agent negotiation in competitive settings. Our results show that SIRIUS enhances multi-agent performance while generating reusable data for self-correction and self-play enhancement in the future.

## 1 Introduction

Multi-agent AI systems powered by large language models (LLMs), where specialized agents collaborate to solve complex tasks, are becoming increasingly successful in real-world applications. Recent work has demonstrated their effectiveness in complex reasoning (Wang et al., 2024; Smit et al., 2024), coding (Wu et al., 2023), drug discovery (Swanson et al., 2024) and ensuring safety via debate (Chern et al., 2024; Irving et al., 2018). These successes arise from specialized agents integrating their distinct capabilities through structured interactions, enabling more effective problem-solving than single agents. Moreover, multi-agent scrutiny acts as a built-in self-correction mechanism, where agents refine and verify each other's outputs. This often outperforms single agent setting, particularly on tasks demanding rigorous reasoning or factual validation. Despite these successes, optimizing multi-agent systems remains a fundamental challenge due to (1) the difficulty of acquiring appropriate training signals for each agent and (2) the sensitivity to multiple moving parts that influence overall performance (Smit et al., 2024). While task-level reward feedback is available, credit assignment across agents remains ambiguous—it is unclear how to attribute success or failure to specific intermediate decisions and reasoning steps made by each LLM agent. This challenge parallels the multi-agent credit assignment problem in reinforcement learning (Foerster et al., 2018). However, in language-based systems, reasoning unfolds through complex and unstructured interactions, making attribution far more difficult than in traditional RL settings with well-defined action spaces.

We present SIRIUS, a framework for learning effective multi-agent behaviors from outcome rewards. Our key insight is that when multiple agents successfully solve a task together, their entire interaction

---

Correspondence to wanjiazh@cs.stanford.edu and jamesz@cs.stanford.edu. Code is available here.

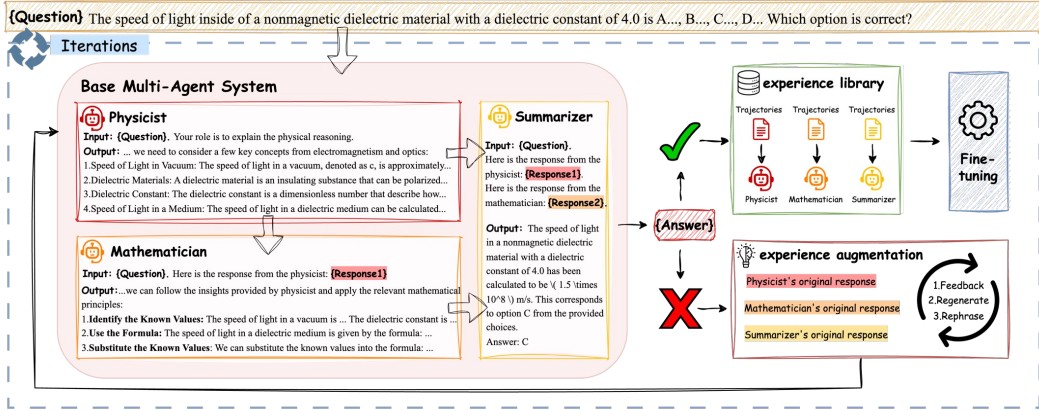

Figure 1: General training pipeline of SIRIUS. Agents solve problems sequentially, storing correct responses for fine-tuning and augmenting incorrect ones through feedback, regeneration, and rephrasing. This iterative process improves performance via reward-based evaluation and supervised fine-tuning. The module colors in the figure correspond to those in Algorithm 1.

trajectory likely contains useful patterns - even if we cannot pinpoint exactly which steps or decisions were crucial for success. Drawing inspiration from recent advances in bootstrapping reasoning capabilities (Zelikman et al., 2022), we collect and learn from successful agent interactions across many tasks, allowing the system to iteratively discover effective collaboration strategies from self-generated data. This approach sidesteps the need for direct supervision of intermediate steps, instead letting agents learn which interaction patterns tend to lead to successful outcomes. For trajectories that result in failed attempts, we perform trajectory augmentation by resampling original attempts with feedback from an additional agent grounded in the ground truth.

Our experiments demonstrate that SIRIUS significantly enhances multi-agent performance across multiple domains. It improves reasoning and biomedical QA accuracy by 2.86% to 21.88%, while also strengthening agent negotiation in competitive scenarios. Beyond these gains, our approach offers a scalable mechanism for self-improvement, enabling agents to iteratively refine their reasoning and collaboration strategies. More broadly, SIRIUS provides a general framework for optimizing multi-agent systems via self-generated synthetic data, offering a principled way to enhance performance without requiring fine-grained human supervision.

## 2 Method

### 2.1 Multi-agent systems with LLMs

We define a multi-agent system by a tuple $\langle \mathcal{S}, \mathcal{A}, \mathcal{T}, \mathcal{R}, \mathcal{N}, \mathcal{G} \rangle$. Here, $\mathcal{N} \triangleq \{A^{(1)}, A^{(2)}, \dots, A^{(N)}\}$ is the set of $N$ agents, each agent $A^{(i)}$ uses a policy $\pi_i$ parameterized by $\theta^{(i)}$. $s \in \mathcal{S}$ is the state of the environment, $\mathbf{a} \in \mathcal{A}$ is the joint actions, and $\mathcal{A}$ is the joint action space. $\mathcal{T} : \mathcal{S} \times \mathcal{A} \to \mathcal{S}$ is the transition function where $\mathcal{T}(s, \mathbf{a})$ yields the next state of the environment given the current state and joint actions $\mathbf{a}$. The environment feedback is modeled via a payoff function $\mathcal{R}_i : \mathcal{S} \times \mathcal{A} \to \mathbb{R}^N$, which provides rewards for each agent $k$ based on the state-action pairs.

The communication structure between agents is modeled as a directed graph $\mathcal{G} = (\mathcal{V}, E, \mathcal{P})$, where $\mathcal{V}$ represents agents, and $E$ defines interaction order.

For each edge $(i, j) \in E$, agent $A^{(j)}$ receives an input derived from the state-action pair $(s, \mathbf{a})$ and the output of agent $A^{(i)}$. This input determines agent $A^{(j)}$'s subsequent action. For each agent $A^{(i)}$ in a topological graph $\mathcal{G}$, its predecessors are the set of agents that influence its output: $\text{Pre}(A^{(i)}) = \{A^{(j)} \mid (A^{(j)}, A^{(i)}) \in \mathcal{G}\}$. Here, $(A^{(j)}, A^{(i)})$ denotes a directed edge in the graph, indicating that the output of agent $A^{(j)}$ directly influences the input of agent $A^{(i)}$.

Throughout this paper, the collection of our agents will be based on language models and the primary environment that we use will be natural language. In particular:

$$a_i \sim \pi_i(\cdot|s_t, \{a_j\}_{A^{(j)} \in \mathrm{Pre}(A^{(i)})}) \quad \forall A^{(i)} \in \mathcal{N}$$
$$\mathbf{a}_t = (a_1, ..., a_N) \tag{1}$$
$$s_{t+1} = \mathcal{T}(s_t, \mathbf{a}_t) = \mathrm{Concat}(s_t, \mathbf{a}_t)$$

where $\pi_i$ denotes the probability distribution of the $i$-th language model, Concat is the concatenation of the previous state and the responses, and we will use $\pi = \{\pi_1, \ldots, \pi_N\}$ to denote the joint policy. Generally, each agent aims to maximize its own reward: $\max_{\pi_i} \mathbb{E}_\pi \left[ \sum_{t=0}^{\infty} R_i(s_t, \mathbf{a}_t) \right]$, where $R_i$ denotes the $i$-th component of the reward vector $\mathcal{R}$ and the expectation is taken under the joint policy $\pi$.

## 2.2 SIRIUS

The training pipeline of the proposed framework, denoted as SIRIUS, is illustrated in Figure 1. SIRIUS adopts a fine-tuning strategy to iteratively improve the policy parameters $\theta^{(n)}$ of each agent $A^{(n)}$ over $T$ iterations. The process is initialized with a dataset $\mathcal{D} = \{(x_i, y_i)\}_{i=1}^D$, where each pair $(x_i, y_i)$ represents a problem and its solution. The core training procedure is outlined in Algorithm 1.

---

**Algorithm 1** SIRIUS

1: **Input:** A group of agents $A^{(1)}, \cdots, A^{(N)}$ An initial dataset of problems $x$ with answer $y$ : $\mathcal{D} = \{(x_i, y_i)\}_{i=1}^D$, total number of fine-tuning Iterations $T$.
2: **Initialize:** Initialize policy parameters $\theta^{(n)}$ for each agent $A^{(n)}$, $k = 1, 2, \ldots, N$.
3: **for** Fine-tuning Iteration t $= 1, \cdots, T$ **do**
4:     $a_i^{(n)} = \mathcal{P}_{\theta_t^{(n)}}(\cdot|x_i, \mathbf{a}_i^{\mathrm{Pre}(A^{(n)})}), k = 1, 2, \ldots, K.$
5:     **for** each agent $k = 1, 2, \ldots, K$ **do**
6:         $\mathcal{C}_t^{(n)} \leftarrow \{(x_i, a_i^{(n)}|i \in [1, D] \wedge R_i(s, a) > \epsilon)\}$
        Good Trajectory Set of Each Agent.
7:         **Augmentation**$(\{(x_i, a_i^{(n)} \wedge R_i(s, a) < \epsilon)\})$
8:     **end for**
9:     $\theta_t^{(n)} \leftarrow$ **Standard SFT on** $\mathcal{C}_t^{(n)}$, $n = 1, \cdots, N$
10: **end for**

---

At each fine-tuning iteration $t$:

- **Action Sampling:** For each agent $A^{(n)}$, an action $a_i^{(n)}$ is sampled from its policy,

$$a_i^{(n)} = \mathcal{P}_{\theta^{(n)}}(\cdot|x_i, \mathbf{a}_i^{\mathrm{Pre}(A^{(n)})}),$$

  conditioned on the input problem $x_i$ and the action set $\mathbf{a}_i^{\mathrm{Pre}(A^{(n)})}$ generated by previous agents. In scenarios involving multiple interaction rounds, such as the Competitive Setting, $\mathbf{a}_i^{\mathrm{Pre}(A^{(n)})}$ includes outputs from all agents in all preceding rounds.
- **Trajectory Evaluation and Augmentation**: The trajectories generated by each agent are evaluated using the payoff function $R(s, \mathbf{a})$. Based on a reward threshold $\epsilon$, high-reward trajectories ($R(s, \mathbf{a}) > \epsilon$) are added to the good trajectory set $\mathcal{C}_t^{(n)}$. Since the tasks are challenging, the good trajectory set tends to be small. To leverage more data for fine-tuning, we propose trajectory augmentation pipeline for each task, detailed in the Appendix C. Specifically, we first generate feedback to refine the agent's original response. The feedback and original response are then combined to prompt the agent to regenerate a new solution, which is then rephrased into a direct problem-solving step. Afterward, we return to the action sampling process to produce the final answer and evaluate it.
- **Fine-Tuning:** The policy parameters $\theta^{(n)}$ are updated via supervised fine-tuning on $\mathcal{C}_t^{(n)}$.

This iterative process ensures that each agent's policy is progressively refined to maximize performance based on the joint system dynamics and reward.

Table 1: Different settings and tasks. In the rows corresponding to Communication Structure, nodes denote agents ($\mathcal{V}$), arrows represent edges ($E$), and color indicates the role of agents.

| Settings | Problem-Solving | | Actor-Critic | Competitive |
|---|---|---|---|---|
| Structure $(\mathcal{V}, E, \mathcal{P})$ |  |  |  |  |
| Tasks | College-Physics College-Chemistry | PubMedQA | PubMedQA | Resource Exchange Seller-Buyer Ultimatum |
| Reward for each agent $R_i$ | Final Output Correctness | | Final Output Correctness | Utility Function Value |

# 3 Multi-agent Settings

In this section, we explore several settings where agents with distinct expertise interact to solve challenging tasks. As shown in Table 1, we systematically analyze different agent configurations.

## 3.1 Problem Solving Settings

**Agents with Specific Expertise.** In this setting, each agent is assigned a domain-specific role to facilitate a structured and efficient problem-solving process. For instance, in the physics and chemistry domains, the problem-solving pipeline begins with a domain expert (e.g., a physicist or chemist) who analyzes the domain-specific problem, followed by a mathematician who formalizes the reasoning with quantitative models, and finally, a summarizer who consolidates the insights into a clear and comprehensive answer. This sequential collaboration ensures that the expertise of each agent is leveraged effectively while maintaining clarity in the solution process.

The sequential dependency between the agents can be described as follows:

$$a_{\text{Phy}} \sim \pi_{\text{Phy}}(\cdot|q), \tag{2}$$
$$a_{\text{Math}} \sim \pi_{\text{Math}}(\cdot|q, a_{\text{Phy}}), \tag{3}$$
$$a_{\text{Sum}} \sim \pi_{\text{Sum}}(\cdot|q, a_{\text{Phy}}, a_{\text{Math}}), \tag{4}$$

where $q$ is the input question, $a_{\text{Phy}}$ is the response generated by the Physicist, $a_{\text{Math}}$ is the response generated by the Mathematician based on both the question and the Physicist's response, $a_{\text{Sum}}$ is the final answer synthesized by the Summarizer using the question, the Physicist's response, and the Mathematician's response.

**Analyze Long Context and Answer Question.** In scenarios involving lengthy and complex contexts, we consider a common two-agent setup: the Context Analyst and the Problem Solver. The Context Analyst's responsibility is to thoroughly examine the context, extract essential information, and provide a concise and accurate summary. The Problem Solver then uses this summary to analyze the question and formulate the final answer. This division of labor not only improves interpretability, but also reduces the cognitive load on each agent.

## 3.2 Actor-Critic Setting

The popular Actor-Critic framework facilitates iterative agent improvement through a feedback loop: the Actor Agent generates solutions while the critic evaluates and refines them, enhancing both the Actor Agent's reasoning and the Critic Agent's error correction capabilities. In practice, we separate judgment and feedback tasks by introducing a Judgment Agent alongside the Critic Agent, where the Judgment Agent classifies the Actor Agent's solutions as correct or incorrect, and for incorrect solutions, the critic provides feedback to guide the Actor Agent in regenerating improved solutions. Reward mechanisms are designed as: the Actor Agent receives rewards for correct solutions, the Judgment Agent for accurate classifications, and the critic for providing actionable feedback that leads to correct regenerations.

## 3.3 Competitive Settings

Competitive scenarios (Bianchi et al., 2024) examine multi-agent interactions under opposing objectives, where agents must balance cooperation and competition to achieve their goals. In this

Table 2: Tasks and setups in the competitive setting. Each task involves two agents with distinct roles, initial resources, and objectives. *Resource Exchange* focuses on maximizing total resources through trade. Ultimatum requires negotiating a split of $100. *Sell&Buy* involves price negotiation for an item. Each task follows a turn-based structure with a fixed maximum number of rounds and ends when an agreement is reached.

| Task | Resource Exchange | | Ultimatum | | Sell&Buy | |
|---|---|---|---|---|---|---|
| Roles | Player 1 | Player 2 | Player 1 | Player 2 | Seller | Buyer |
| Initial resources | 25Xs, 5Ys | 5Xs, 25Ys | $ 100 | 0 | 1X | 100 ZUPs |
| Goal | Maximize total resources | | Negotiate a split | | Maximize price | Minimize price |
| Utility | Xs + Ys | Xs + Ys | Split amount-50 | Split amount-50 | Selling price - 50 | 50-Selling price |
| Ending condition | When either player accepts | | When either player accepts | | When either player accepts | |
| Max. # of turns | 8 rounds of interaction | | 8 rounds of interaction | | 10 rounds of interaction | |

framework, two agent roles are defined: **Player 1** and **Player 2**. Each player is initialized with a specific amount of resources, which evolve over the course of the game based on their interactions. The game progresses as a sequence of moves, resulting in a trajectory of states:

$$\text{Player 1 Trajectory: } x_0^{\text{player1}}, x_1^{\text{player1}}, \cdots, x_T^{\text{player1}}$$
$$\text{Player 2 Trajectory: } x_0^{\text{player2}}, x_1^{\text{player2}}, \cdots, x_T^{\text{player2}}$$
(5)

The sequence captures the evolution of game states as players compete at each timestep $t = 0, 1, \ldots, T$, ultimately determining a winner and a loser. Our goal is to optimize each player's policy to maximize its own expected reward based on trajectory data and role-specific context. This can be formulated as:

$$\max \sum_{i=1}^{T} P_\theta(x_i^{\text{player1}} | x_{0:i-1}^{\text{player1}}, x_{0:i-1}^{\text{player2}})$$
(6)

where Player 1 optimizes its policy based on the historical trajectory of both itself and Player 2, and similarly for Player 2. We explore three distinct competitive settings, all of which unfold over multiple rounds:

**Resource Exchange Scenario.** In this scenario, agents engage in a simulated environment where they exchange resources to maximize their individual utility.

**Seller and Buyer Scenario.** This setting models economic interactions where one agent assumes the role of a seller and another the role of a buyer. The agents negotiate prices and terms to complete transactions, testing their ability to strategize under asymmetric setting.

**Multi-Turn Ultimatum Game.** The Multi-Turn Ultimatum Game explores scenarios of fairness, cooperation, and negotiation over multiple rounds. One agent proposes a division of a resource, and the other agent decides whether to accept or reject it.

# 4 Experiments

## 4.1 Baseline

We compare our SIRIUS against the following baselines:

**Single-Agent** utilizes a single language model to process input and generate responses.

**STaR** (Zelikman et al., 2022), the Self-Taught Reasoner, focuses on enhancing the reasoning capabilities of a single agent by iteratively training it to improve its step-by-step reasoning through self-supervised fine-tuning.

**Prompt Multi-Agent System (CoMM)** (Chen et al., 2024a) introduces a training-free, multi-agent collaborative framework where agents interact and share information to solve tasks collectively.

**TextGrad** (Yuksekgonul et al., 2024) optimizes prompts for each agent in a multi-agent system by backpropagating natural language feedback through each interaction.

**DSPy (MIPROv2)** (Khattab et al., 2023)is a prompt optimizer that jointly optimizes instructions and few-shot examples.

Table 3: Evaluation results of the proposed method and baselines on accuracy(%). Best results are in **bold** numbers and second-best results are in underline numbers.

| Model | Method | College Physics | College Chemistry | PubMedQA |
|---|---|---|---|---|
| GPT-3.5-turbo | **Single-Agent** | $25.55 \pm 1.08$ | $40.00 \pm 3.95$ | $57.53 \pm 0.99$ |
| | **STaR** | $30.84 \pm 0.93$ | $45.64 \pm 1.72$ | $64.20 \pm 0.53$ |
| | **COMM** | $28.97 \pm 1.62$ | $\underline{47.69 \pm 3.95}$ | $\underline{72.27 \pm 0.81}$ |
| | **TextGrad** | $\underline{32.09 \pm 1.08}$ | $41.54 \pm 0.00$ | NA |
| | **Dspy** | $29.91 \pm 1.87$ | $39.49 \pm 2.45$ | $52.67 \pm 1.29$ |
| | SIRIUS | $\mathbf{33.96 \pm 1.43}$ | $\mathbf{55.90 \pm 3.11}$ | $\mathbf{75.33 \pm 0.70}$ |
| GPT-4o-mini | **Single-Agent** | $38.63 \pm 1.95$ | $40.00 \pm 2.59$ | $63.87 \pm 0.12$ |
| | **STaR** | $42.99 \pm 0.93$ | $48.21 \pm 2.28$ | $65.93 \pm 2.83$ |
| | **COMM** | $42.37 \pm 1.43$ | $49.23 \pm 1.49$ | $\underline{70.27 \pm 0.42}$ |
| | **TextGrad** | $39.25 \pm 3.37$ | $\underline{49.74 \pm 7.36}$ | $66.20 \pm 2.11$ |
| | **Dspy** | $\underline{46.11 \pm 1.95}$ | $45.65 \pm 2.35$ | $55.53 \pm 0.31$ |
| | SIRIUS | $\mathbf{47.35 \pm 1.95}$ | $\mathbf{56.41 \pm 3.11}$ | $\mathbf{73.67 \pm 0.31}$ |
| Llama-3.2-3B-instruct | **Single-Agent** | $26.79 \pm 1.95$ | $34.46 \pm 2.28$ | $54.07 \pm 0.31$ |
| | **STaR** | $27.41 \pm 1.95$ | $\underline{38.46 \pm 1.49}$ | $54.93 \pm 0.12$ |
| | **COMM** | $\underline{27.73 \pm 1.43}$ | $37.44 \pm 3.11$ | $\underline{65.80 \pm 0.87}$ |
| | SIRIUS | $\mathbf{29.60 \pm 1.43}$ | $\mathbf{42.56 \pm 2.28}$ | $\mathbf{68.27 \pm 1.03}$ |

## 4.2 Setup and Datasets

**Backbone Model.** For a fair comparison, we use gpt-3.5-turbo-0125 and gpt-4o-mini-2024-07-18 as the backbone model, and set the temperature to 0 in all our experiments. We use OpenAI's Fine-tuning API for supervised fine-tuning.

**College Physics/Chemistry.** These two datasets are constructed by combining questions from Massive Multitask Language Understanding (MMLU) (Hendrycks et al., 2020), Graduate-Level Google-Proof Q&A (GPQA) (Rein et al., 2023), and Theorem-Driven Question Answering (TheoremQA) (Chen et al., 2023). It focuses on college-level physics problems, which remain difficult and demonstrate room for improvement in performance with large language models. We split the dataset into training and test sets, with the detailed data distribution provided in Appendix D.

**PubMedQA.** This is a biomedical question-answering dataset comprising 1000 open-domain questions (Jin et al., 2019), each paired with context from PubMed abstracts and corresponding answers. It focuses on research-driven queries, requiring domain-specific understanding and reasoning over scientific texts. We follow the original split of the dataset for training (500) and testing (500) sets.

## 4.3 Experimental Result of Problem Solving Setting

### 4.3.1 Main Result

Table 3 presents a performance comparison of various models and methods under the Problem Solving Setting. We observe that the prompted Multi-Agent System (COMM) generally improves performance, as agent collaboration enhances the ability to solve complex problems. STaR outperforms the base Single-Agent, indicating that fine-tuning contributes to improved performance. For smaller and weaker models, and in scenarios with long context lengths such as PubMedQA, TextGrad faces significant challenges in instruction-following during optimization. TextGrad (GPT-3.5-turbo) could not be applied to PubMedQA as its optimizer failed to parse instructions due to the model's limited capability and the excessive context length of the problem. Similarly, TextGrad (GPT-4o-mini) struggles to generate answers in the required format, requiring manual extraction of answers. Our proposed method, SIRIUS, consistently outperforms across all tasks. By decomposing tasks into manageable sub-tasks assigned to agents and, crucially, fine-tuning each agent to specialize in its designated task, SIRIUS maximizes the effectiveness of collaboration, ensuring a more coordinated and efficient overall performance.

### 4.3.2 Ablation Experiments

To evaluate the contributions of various components in SIRIUS, we conducted a series of ablation experiments. Each experiment was designed to answer a key question about the effectiveness of

the multi-agent system. All ablations were performed on representative tasks within the Problem Solving Setting (PubMedQA) to ensure consistency in evaluation as shown in Table 4.

**Does mixing SIRIUS with a base agent degrade performance?** To understand the benefits of a jointly optimizing a collaborative multi-agent system, we first train all the agents together using SIRIUS. Then we replaced one SIRIUS agent with the original base agent—either SIRIUS Analyst + base Solver or base Analyst + SIRIUS Solver. This substitution hurts performance, demonstrating benefits from joint multi-agent optimization compared to optimizing a single agent.

**Should we fine-tune different LLMs for different roles, or optimize one LLM for all roles?** We explored whether a single LLM fine-tuned on the combined training data of multiple roles could match the performance of separate role-specific models. The results showed a notable performance decline, highlighting that different roles require specialized adaptation and that a shared model struggles to effectively generalize across distinct agent functions.

**How useful is experience augmentation?** To assess the impact of experience augmentation, we removed the augmentation module while keeping the rest of the pipeline unchanged. Data augmentation introduces more diverse and challenging experiences as training data, enhancing the model's capability; therefore, omitting the augmentation module could negatively impact performance.

**Does additional fine-tuning improve performance?** We investigated whether increasing the number of fine-tuning iterations leads to further performance gains. Each iteration follows the full optimization pipeline illustrated in Figure 1, the previously fine-tuned SIRIUS is used to generate a new experience library, which is then used to further fine-tune the base model. As expected, an additional iteration yielded marginal performance gains, suggesting that the model can benefit from extended training.

Table 4: Ablation results on PubMedQA.

| Model | method | PubMed |
|---|---|---|
| GPT-3.5-turbo | SIRIUS | 74.20 |
| | SIRIUS + Base | 72.00 |
| | Base + SIRIUS | 73.20 |
| | FT on One Base LLM | 70.40 |
| | SIRIUS w/o Aug. | 73.40 |
| | Additional FT Itr | 75.00 |
| GPT-4o-mini | SIRIUS | 73.40 |
| | SIRIUS + Base | 72.80 |
| | Base + SIRIUS | 71.60 |
| | FT on One Base LLM | 72.00 |
| | SIRIUS w/o Aug. | 72.20 |
| | Additional FT Itr | 73.60 |

## 4.4 Experimental Result of Actor-Critic Setting

Table 5 presents a performance comparison of various models, methods, and ablations under the Actor-Critic Setting on PubMedQA. As mentioned in Section 3.2, the Actor Agent first generates a solution, which is then evaluated by the Judgment Agent to determine its correctness. For solutions deemed incorrect by the Judgment Agent, the Critic Agent analyzes the original solution and provides feedback without access to the correct answer. The Actor Agent then regenerates the solution based on this feedback.

A key challenge in this setting is the Judgment Agent's limited ability to differentiate between correct and incorrect solutions leading to two potential issues: (1) correct solutions may be mistakenly judged as incorrect and potentially modified into incorrect ones during the feedback and regeneration stages; (2) incorrect solutions may be judged as correct, failing to receive the necessary corrections. We report TP (True Positive) Accuracy as the ratio of solutions both correctly generated by the Actor and accurately validated by the Judgment Agent, while Overall Accuracy measures the total correct solutions after regeneration, accounting for the combined contributions of all agents.

We evaluate our method against two representative baselines: (1) Self-Correct, where Actor-generated solutions are refined through direct feedback-guided regeneration, and (2) Prompt, which exclusively employs prompting strategies to coordinate Actor-Judgment-Critic interactions without optimization mechanisms. A critical limitation observed in the Self-Correct framework is its significantly lower TP accuracy. This issue arises from its feedback mechanism, which modifies all generated responses with high probability, potentially leading to erroneous modifications of the initially correct solution. This is a common issue with using out-of-the-box LLMs for self-correction with no specialized training (Kumar et al., 2024).

Table 5: Evaluation results of the proposed method and baselines on accuracy(%).

| Model | GPT-3.5-Turbo | | GPT-4o-mini | |
|---|---|---|---|---|
| **Method** | TP Accuracy | Overall Accuracy | TP Accuracy | Overall Accuracy |
| Self-Correct | 11.80 | 16.40 | 24.60 | 28.80 |
| Prompt | 18.40 | 47.60 | 51.60 | 58.20 |
| SIRIUS | **35.00** | **50.60** | **59.80** | **66.80** |
| | **Ablation Study** | | | |
| SIRIUS + BASE Actor Agent | 34.20 | 49.00 | 49.60 | 54.40 |
| SIRIUS + BASE Judgment Agent | 20.20 | 40.20 | 53.00 | 59.40 |
| SIRIUS + BASE Critic Agent | 35.00 | 50.40 | 59.80 | 64.20 |
| FT on One Base LLM | 33.80 | 43.60 | 56.00 | 59.60 |

Comparing GPT-3.5-Turbo and GPT-4o-mini, we also find that GPT-3.5-Turbo struggles more with misjudging correct answers as incorrect, leading to a severe drop in TP Accuracy. Our method, SIR-IUS, achieves a notable improvement in TP Accuracy, highlighting the Judgment Agent's enhanced ability to assess whether a response requires modification. The overall higher accuracy underscores the effectiveness of SIRIUS's framework, where fine-tuning enhances each agent's task-specific capabilities, and the collaboration of Judgment, Critic, and Actor Agents ensures appropriate revision of incorrect responses while minimizing unnecessary changes to correct answers.

The ablation study further underscores the contribution of each agent in SIRIUS. Fine-tuning only a single base LLM leads to a performance drop, highlighting the necessity of specialized agent roles and joint optimization. Notably, replacing the Judgment Agent with a baseline version significantly reduces TP Accuracy, reinforcing its essential role in filtering correct responses before feedback is applied.

## 4.5 Experimental Result of Competitive Settings

To analyze the effect of training in the competitive setting, we study the performance of agents in scenarios where one player initially had a higher probability of winning, referred to as the "winning player," while the other player was at a disadvantage, called the "losing player." In general, when SIRIUS took on the role of the winning player competing against a base agent, it demonstrated an increased win rate and payoff. Additionally, when SIRIUS played the role of the losing player, it experienced fewer losses. Similarly, for both GPT-3.5 and GPT-4o-mini when they compete with each other, SIRIUS-GPT-3.5 and SIRIUS-GPT-4o-mini both demonstrate improved performance.

### 4.5.1 Resource Exchange

The win rates and average payoffs for the Resource Exchange game are presented in Figure 2. Overall, the agent going second tends to beat the first agent. Furthermore, the fine-tuned SIRIUS demonstrates a significant improvement in both the win rate and payoff for the current player. To evaluate the generalization capability of our approach, we conducted additional experiments with models fine-tuned on games featuring Initial Resource configurations of 25Xs + 5Ys and 5Xs + 25Ys, and then tested them on games with different Initial Resource configurations (35Xs + 15Ys and 15Xs + 35Ys). As demonstrated in Figure 5, SIRIUS

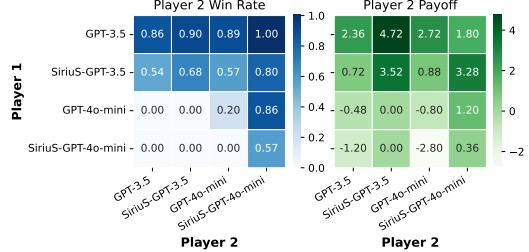

Figure 2: Resource Exchange Game: Player 1 (25Xs + 5Ys), Player 2 (5Xs + 25Ys). Win Rate in decisive games and Payoff in all games. We show Player 2 Win rate/payoff in all cells.

maintains notable improvements in the new Initial Resource configurations, effectively validating the generalizability of our proposed pipeline.

### 4.5.2 Multi-Turn Ultimatum

In this setting, Player 1 consistently dominates the game. As shown in the Figure 3 , SIRIUS fine-tuned Player 1 effectively secure a higher share of the split. Generalization experiments show that

SIRIUS Player 1 trained in the Resource = 100 setting maintains utility gains in the new Resource = 1000 setting (Figure 7).

### 4.5.3 Buyer-Seller

In this setting, sellers are willing to sell when the price exceeds 40, while buyers are willing to buy when the price is below 60. We plot the final selling price as shown in Figure 4. Notably, it is consistently below 50 for most buyer-seller pairs, indicating that the LLM agent performs better as a buyer than as a seller. After fine-tuning, SIRIUS as a seller shows significant improvement, consistently selling at 50, resulting in a tie with the buyer. To test the generalization capability and ensure the seller is not overfitting to a price of 50, we adjusted the initial configuration to 30 and 70. Figure 6 shows that the SIRIUS seller trained in the previous setup still demonstrates significant improvement.

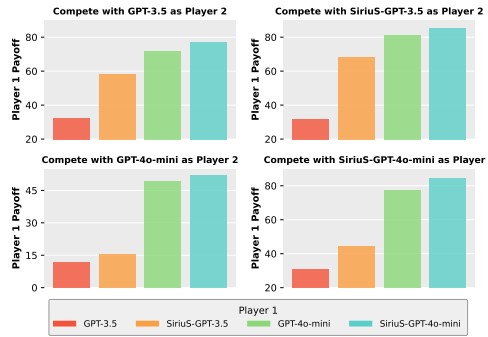 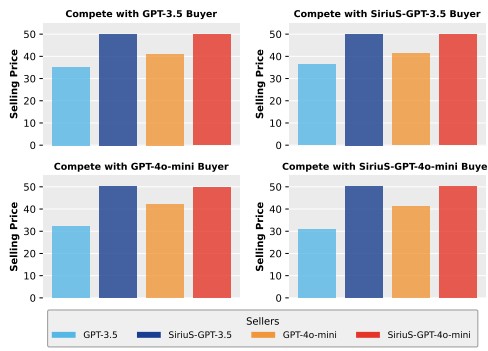

Figure 3: Player 1's payoff in the Ultimatum game with an initial resource of 100. SIRIUS as Player 1 secures a higher share.

Figure 4: Final Selling Price for a Seller&Buyer with object valuations of 40 and 60. A higher number means a greater payoff for Seller.

## 5 Related Work

**Enhancing Reasoning in Single-Agent Systems.** Building on the reasoning capabilities of state-of-the-art foundation models (Schulman et al., 2022; OpenAI, 2023; Liu et al., 2024), recent research explores approaches beyond scaling model parameters. Chain-of-Thought (Wei et al., 2022) enhances reasoning through step-by-step inference, while Tree of Thoughts (Yao et al., 2024), Graph of Thought (Besta et al., 2024), and Program of Thoughts (Chen et al., 2022) structure reasoning as tree searches with backtracking. Reasoning with Planning (RAP) (Hao et al., 2023) incorporates explicit planning, and Reflexion (Shinn et al., 2024) enables self-evaluation and refinement. Wu et al. (2024) introduce contrastive reasoning for instruction generation, while TextGrad (Yuksek-gonul et al., 2024) applies gradient-based optimization to refine outputs. These methods enhance reasoning through structured decomposition, search, and planning.

**Self-improvement.** Self-improving models (Huang et al., 2022; Yu et al., 2023; Yuan et al., 2024; Zhang et al., 2024; Welleck et al., 2022) have garnered increasing attention for their potential to enhance reasoning capabilities through iterative feedback and refinement. Several studies (Zelikman et al., 2022; Li et al., 2024a; Pang et al., 2024; Lee et al., 2024)employ bootstrapping strategies by leveraging self-generated rationales, while others (Yuan et al., 2024; Chen et al., 2024b; Guo et al., 2025) introduces a self-refinement mechanism through reinforcement learning.

**Multi-Agent Systems with LLMs. Multi-Agent Systems with LLMs.** Recent advancements in multi-agent systems (Smit et al., 2024; Guo et al., 2024; Li et al., 2024b; Han et al., 2024) highlight the potential of large language models in tackling complex tasks. Society of Minds (Du et al., 2023) enables agents to exchange answers, fostering collaboration. Mixture-of-Agents (Wang et al., 2024) employs a layered architecture where agents refine responses based on prior outputs. CoMM (Chen et al., 2024a) enhances problem-solving through structured communication and role division. Multi-Persona (Liang et al., 2023) encourages diverse agent behaviors by assigning distinct personas. ChatEval (Chan et al., 2023) explores different multi-agent debate strategies for interaction and response management. Building on advances in multi-agent systems, recent work has explored fine-tuning with independently specialized agents that interact to generate diverse reasoning chains (Subramaniam et al., 2025). Unlike these approaches, our method prioritizes collab-

orative optimization through a shared experience library, enabling agents to collectively learn from and refine successful reasoning trajectories.

## 6 Conclusions

We introduced SIRIUS, a framework for optimizing multi-agent LLM systems by learning from successful interactions and augmenting failed trajectories with feedback. Our approach enables agents to refine collaboration strategies without explicit supervision. Experiments show that SIRIUS significantly improves performance across college-level reasoning, biomedical QA, and negotiation tasks. More broadly, our work provides a scalable mechanism for multi-agent self-improvement, offering a principled approach to optimizing collaborative AI systems.

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

# NeurIPS Paper Checklist

1. **Claims**

   Question: Do the main claims made in the abstract and introduction accurately reflect the paper's contributions and scope?

   Answer: [Yes]

   Justification: The abstract claims that SIRIUS is a self-improving, reasoning-driven optimization framework for multi-agent systems that constructs an experience library of high-quality reasoning trajectories. It also claims performance boosts of 2.86% to 21.88% on reasoning and biomedical QA, and enhanced agent negotiation. The introduction reiterates these points, highlighting the challenges in optimizing multi-agent systems and how SIRIUS addresses them through learning from successful interactions and augmenting unsuccessful ones. The subsequent sections (Methods, Experiments) detail the SIRIUS framework, the construction of the experience library, the augmentation process, and present experimental results that support these claims with specific percentage improvements on various tasks (e.g., Table 3 shows improvements on College Physics, College Chemistry, and PubMedQA).

   Guidelines:

   - The answer NA means that the abstract and introduction do not include the claims made in the paper.
   - The abstract and/or introduction should clearly state the claims made, including the contributions made in the paper and important assumptions and limitations. A No or NA answer to this question will not be perceived well by the reviewers.
   - The claims made should match theoretical and experimental results, and reflect how much the results can be expected to generalize to other settings.
   - It is fine to include aspirational goals as motivation as long as it is clear that these goals are not attained by the paper.

2. **Limitations**

   Question: Does the paper discuss the limitations of the work performed by the authors?

   Answer: [Yes]

   Justification: The paper includes a dedicated "Limitations" section (Appendix B). This section discusses dependence on backbone LLM capabilities, the ongoing challenge of multi-agent credit assignment, and the complexity of designing optimal agent interaction protocols.

   Guidelines:

   - The answer NA means that the paper has no limitation while the answer No means that the paper has limitations, but those are not discussed in the paper.
   - The authors are encouraged to create a separate "Limitations" section in their paper.
   - The paper should point out any strong assumptions and how robust the results are to violations of these assumptions (e.g., independence assumptions, noiseless settings, model well-specification, asymptotic approximations only holding locally). The authors should reflect on how these assumptions might be violated in practice and what the implications would be.
   - The authors should reflect on the scope of the claims made, e.g., if the approach was only tested on a few datasets or with a few runs. In general, empirical results often depend on implicit assumptions, which should be articulated.
   - The authors should reflect on the factors that influence the performance of the approach. For example, a facial recognition algorithm may perform poorly when image resolution is low or images are taken in low lighting. Or a speech-to-text system might not be used reliably to provide closed captions for online lectures because it fails to handle technical jargon.
   - The authors should discuss the computational efficiency of the proposed algorithms and how they scale with dataset size.

- If applicable, the authors should discuss possible limitations of their approach to address problems of privacy and fairness.
- While the authors might fear that complete honesty about limitations might be used by reviewers as grounds for rejection, a worse outcome might be that reviewers discover limitations that aren't acknowledged in the paper. The authors should use their best judgment and recognize that individual actions in favor of transparency play an important role in developing norms that preserve the integrity of the community. Reviewers will be specifically instructed to not penalize honesty concerning limitations.

3. **Theory assumptions and proofs**

   Question: For each theoretical result, does the paper provide the full set of assumptions and a complete (and correct) proof?

   Answer: [NA]

   Justification: The paper primarily presents an empirical framework and experimental results. It defines the multi-agent system formally (Section 2.1) but does not focus on deriving new theoretical results with formal proofs.

   Guidelines:

   - The answer NA means that the paper does not include theoretical results.
   - All the theorems, formulas, and proofs in the paper should be numbered and cross-referenced.
   - All assumptions should be clearly stated or referenced in the statement of any theorems.
   - The proofs can either appear in the main paper or the supplemental material, but if they appear in the supplemental material, the authors are encouraged to provide a short proof sketch to provide intuition.
   - Inversely, any informal proof provided in the core of the paper should be complemented by formal proofs provided in appendix or supplemental material.
   - Theorems and Lemmas that the proof relies upon should be properly referenced.

4. **Experimental result reproducibility**

   Question: Does the paper fully disclose all the information needed to reproduce the main experimental results of the paper to the extent that it affects the main claims and/or conclusions of the paper (regardless of whether the code and data are provided or not)?

   Answer: [Yes]

   Justification: The paper provides details on the backbone models used (gpt-3.5-turbo-0125 and gpt-40-mini-2024-07-18), temperature setting, and the use of OpenAI's Fine-tuning API (Section 4.2). It describes the datasets used (College Physics/Chemistry, PubMedQA) and their splits (Section 4.2, Appendix D). The different multi-agent settings (Problem Solving, Actor-Critic, Competitive) and their agent configurations are detailed (Section 3, Table 1, Table 2). Algorithm 1 outlines the SIRIUS training procedure, and Algorithm 2 in the appendix provides more detail. Appendix F provides extensive details on the prompts used for various agents and tasks. This level of detail should allow others to reproduce the main experimental results.

   Guidelines:

   - The answer NA means that the paper does not include experiments.
   - If the paper includes experiments, a No answer to this question will not be perceived well by the reviewers: Making the paper reproducible is important, regardless of whether the code and data are provided or not.
   - If the contribution is a dataset and/or model, the authors should describe the steps taken to make their results reproducible or verifiable.
   - Depending on the contribution, reproducibility can be accomplished in various ways. For example, if the contribution is a novel architecture, describing the architecture fully might suffice, or if the contribution is a specific model and empirical evaluation, it may be necessary to either make it possible for others to replicate the model with the same dataset, or provide access to the model. In general. releasing code and data

is often one good way to accomplish this, but reproducibility can also be provided via detailed instructions for how to replicate the results, access to a hosted model (e.g., in the case of a large language model), releasing of a model checkpoint, or other means that are appropriate to the research performed.

- While NeurIPS does not require releasing code, the conference does require all submissions to provide some reasonable avenue for reproducibility, which may depend on the nature of the contribution. For example
  (a) If the contribution is primarily a new algorithm, the paper should make it clear how to reproduce that algorithm.
  (b) If the contribution is primarily a new model architecture, the paper should describe the architecture clearly and fully.
  (c) If the contribution is a new model (e.g., a large language model), then there should either be a way to access this model for reproducing the results or a way to reproduce the model (e.g., with an open-source dataset or instructions for how to construct the dataset).
  (d) We recognize that reproducibility may be tricky in some cases, in which case authors are welcome to describe the particular way they provide for reproducibility. In the case of closed-source models, it may be that access to the model is limited in some way (e.g., to registered users), but it should be possible for other researchers to have some path to reproducing or verifying the results.

5. **Open access to data and code**

   Question: Does the paper provide open access to the data and code, with sufficient instructions to faithfully reproduce the main experimental results, as described in supplemental material?

   Answer: [Yes]

   Justification: Please check the code and data details in the supplementary material.

   Guidelines:

   - The answer NA means that paper does not include experiments requiring code.
   - Please see the NeurIPS code and data submission guidelines (`https://nips.cc/public/guides/CodeSubmissionPolicy`) for more details.
   - While we encourage the release of code and data, we understand that this might not be possible, so "No" is an acceptable answer. Papers cannot be rejected simply for not including code, unless this is central to the contribution (e.g., for a new open-source benchmark).
   - The instructions should contain the exact command and environment needed to run to reproduce the results. See the NeurIPS code and data submission guidelines (`https://nips.cc/public/guides/CodeSubmissionPolicy`) for more details.
   - The authors should provide instructions on data access and preparation, including how to access the raw data, preprocessed data, intermediate data, and generated data, etc.
   - The authors should provide scripts to reproduce all experimental results for the new proposed method and baselines. If only a subset of experiments are reproducible, they should state which ones are omitted from the script and why.
   - At submission time, to preserve anonymity, the authors should release anonymized versions (if applicable).
   - Providing as much information as possible in supplemental material (appended to the paper) is recommended, but including URLs to data and code is permitted.

6. **Experimental setting/details**

   Question: Does the paper specify all the training and test details (e.g., data splits, hyperparameters, how they were chosen, type of optimizer, etc.) necessary to understand the results?

   Answer: [Yes]

   Justification: The paper specifies backbone models, temperature, and the use of OpenAI's Fine-tuning API (Section 4.2). Data splits for College Physics/Chemistry and PubMedQA

are provided (Section 4.2, Appendix D). The number of fine-tuning iterations (T) is mentioned as an input to Algorithm 1, and additional fine-tuning iterations are explored in ablation studies. Specific agent roles and their interaction structures are defined for different settings. Appendix F details the prompts. While the choice of every hyperparameter (e.g., learning rate for SFT, reward threshold $\epsilon$) isn't exhaustively detailed, sufficient information is present to understand the setup.

Guidelines:

- The answer NA means that the paper does not include experiments.
- The experimental setting should be presented in the core of the paper to a level of detail that is necessary to appreciate the results and make sense of them.
- The full details can be provided either with the code, in appendix, or as supplemental material.

7. **Experiment statistical significance**

Question: Does the paper report error bars suitably and correctly defined or other appropriate information about the statistical significance of the experiments?

Answer: [Yes]

Justification: Table 3 and Table 5, which report the main quantitative results, include values with a "±" notation (e.g., "$25.55 \pm 1.08$"). This suggests reporting of mean and standard deviation or standard error, indicating variability across runs or a similar measure of statistical significance. The paper does not explicitly state how these were calculated (e.g., number of runs for averaging) but their presence addresses the reporting of variability.

Guidelines:

- The answer NA means that the paper does not include experiments.
- The authors should answer "Yes" if the results are accompanied by error bars, confidence intervals, or statistical significance tests, at least for the experiments that support the main claims of the paper.
- The factors of variability that the error bars are capturing should be clearly stated (for example, train/test split, initialization, random drawing of some parameter, or overall run with given experimental conditions).
- The method for calculating the error bars should be explained (closed form formula, call to a library function, bootstrap, etc.)
- The assumptions made should be given (e.g., Normally distributed errors).
- It should be clear whether the error bar is the standard deviation or the standard error of the mean.
- It is OK to report 1-sigma error bars, but one should state it. The authors should preferably report a 2-sigma error bar than state that they have a 96% CI, if the hypothesis of Normality of errors is not verified.
- For asymmetric distributions, the authors should be careful not to show in tables or figures symmetric error bars that would yield results that are out of range (e.g. negative error rates).
- If error bars are reported in tables or plots, The authors should explain in the text how they were calculated and reference the corresponding figures or tables in the text.

8. **Experiments compute resources**

Question: For each experiment, does the paper provide sufficient information on the computer resources (type of compute workers, memory, time of execution) needed to reproduce the experiments?

Answer: [No]

Justification: The paper mentions the LLMs used (gpt-3.5-turbo, gpt-4o-mini) and the use of OpenAI's Fine-tuning API. However, specific details about the compute resources (e.g., number of GPUs, type of GPUs, memory, execution time for fine-tuning or experiments) is unclear.

Guidelines:

- The answer NA means that the paper does not include experiments.

- The paper should indicate the type of compute workers CPU or GPU, internal cluster, or cloud provider, including relevant memory and storage.
- The paper should provide the amount of compute required for each of the individual experimental runs as well as estimate the total compute.
- The paper should disclose whether the full research project required more compute than the experiments reported in the paper (e.g., preliminary or failed experiments that didn't make it into the paper).

9. **Code of ethics**

   Question: Does the research conducted in the paper conform, in every respect, with the NeurIPS Code of Ethics https://neurips.cc/public/EthicsGuidelines?

   Answer: [Yes]

   Justification: The research involves LLMs and their fine-tuning for various tasks. The paper discusses broader impacts, including potential negative societal impacts and the inheritance of biases from backbone LLMs (Appendix A). It uses established datasets and platforms. There's no indication of ethical violations like plagiarism, data fabrication, or harm to human subjects. The responsible disclosure of potential misuse and biases aligns with ethical considerations.

   Guidelines:

   - The answer NA means that the authors have not reviewed the NeurIPS Code of Ethics.
   - If the authors answer No, they should explain the special circumstances that require a deviation from the Code of Ethics.
   - The authors should make sure to preserve anonymity (e.g., if there is a special consideration due to laws or regulations in their jurisdiction).

10. **Broader impacts**

   Question: Does the paper discuss both potential positive societal impacts and negative societal impacts of the work performed?

   Answer: [Yes]

   Justification: Appendix A is titled "Broader Impact" and explicitly discusses both positive societal impacts (e.g., applications in biomedical research, scientific discovery) and potential negative societal impacts (e.g., misuse in competitive negotiation, exploitation, unfair advantages, inherited biases from LLMs, alignment with human values).

   Guidelines:

   - The answer NA means that there is no societal impact of the work performed.
   - If the authors answer NA or No, they should explain why their work has no societal impact or why the paper does not address societal impact.
   - Examples of negative societal impacts include potential malicious or unintended uses (e.g., disinformation, generating fake profiles, surveillance), fairness considerations (e.g., deployment of technologies that could make decisions that unfairly impact specific groups), privacy considerations, and security considerations.
   - The conference expects that many papers will be foundational research and not tied to particular applications, let alone deployments. However, if there is a direct path to any negative applications, the authors should point it out. For example, it is legitimate to point out that an improvement in the quality of generative models could be used to generate deepfakes for disinformation. On the other hand, it is not needed to point out that a generic algorithm for optimizing neural networks could enable people to train models that generate Deepfakes faster.
   - The authors should consider possible harms that could arise when the technology is being used as intended and functioning correctly, harms that could arise when the technology is being used as intended but gives incorrect results, and harms following from (intentional or unintentional) misuse of the technology.
   - If there are negative societal impacts, the authors could also discuss possible mitigation strategies (e.g., gated release of models, providing defenses in addition to attacks, mechanisms for monitoring misuse, mechanisms to monitor how a system learns from feedback over time, improving the efficiency and accessibility of ML).

11. **Safeguards**

Question: Does the paper describe safeguards that have been put in place for responsible release of data or models that have a high risk for misuse (e.g., pretrained language models, image generators, or scraped datasets)?

Answer: [NA]

Justification: The paper focuses on a framework (SIRIUS) and its application using existing LLMs provided by OpenAI. It does not appear to be releasing new large-scale pretrained models or datasets that would pose a high risk for misuse beyond the risks inherent in the underlying LLMs themselves. The discussion of negative impacts and inherited biases is present, but specific safeguards for a *newly released asset* are not applicable as no such new high-risk asset seems to be released by this paper itself.

Guidelines:

- The answer NA means that the paper poses no such risks.
- Released models that have a high risk for misuse or dual-use should be released with necessary safeguards to allow for controlled use of the model, for example by requiring that users adhere to usage guidelines or restrictions to access the model or implementing safety filters.
- Datasets that have been scraped from the Internet could pose safety risks. The authors should describe how they avoided releasing unsafe images.
- We recognize that providing effective safeguards is challenging, and many papers do not require this, but we encourage authors to take this into account and make a best faith effort.

12. **Licenses for existing assets**

Question: Are the creators or original owners of assets (e.g., code, data, models), used in the paper, properly credited and are the license and terms of use explicitly mentioned and properly respected?

Answer: [Yes]

Justification: The paper properly credits the creators of existing datasets through citations (e.g., MMLU - Hendrycks et al., 2020; GPQA - Rein et al., 2023; TheoremQA - Chen et al., 2023; PubMedQA - Jin et al., 2019).

Guidelines:

- The answer NA means that the paper does not use existing assets.
- The authors should cite the original paper that produced the code package or dataset.
- The authors should state which version of the asset is used and, if possible, include a URL.
- The name of the license (e.g., CC-BY 4.0) should be included for each asset.
- For scraped data from a particular source (e.g., website), the copyright and terms of service of that source should be provided.
- If assets are released, the license, copyright information, and terms of use in the package should be provided. For popular datasets, `paperswithcode.com/datasets` has curated licenses for some datasets. Their licensing guide can help determine the license of a dataset.
- For existing datasets that are re-packaged, both the original license and the license of the derived asset (if it has changed) should be provided.
- If this information is not available online, the authors are encouraged to reach out to the asset's creators.

13. **New assets**

Question: Are new assets introduced in the paper well documented and is the documentation provided alongside the assets?

Answer: [NA]

Justification: The primary contribution is the SIRIUS framework. While new fine-tuned models are generated as part of the experiments, the paper doesn't explicitly state these models are being released as new assets. The combined datasets (College Physics/Chemistry) are derived from existing ones. If the framework's code or the specific fine-tuned model weights were to be released, then this question would apply more directly. Based on the text, it doesn't seem to introduce new, distributable assets in that sense.

Guidelines:

- The answer NA means that the paper does not release new assets.
- Researchers should communicate the details of the dataset/code/model as part of their submissions via structured templates. This includes details about training, license, limitations, etc.
- The paper should discuss whether and how consent was obtained from people whose asset is used.
- At submission time, remember to anonymize your assets (if applicable). You can either create an anonymized URL or include an anonymized zip file.

14. **Crowdsourcing and research with human subjects**

Question: For crowdsourcing experiments and research with human subjects, does the paper include the full text of instructions given to participants and screenshots, if applicable, as well as details about compensation (if any)?

Answer: [NA]

Justification: The paper does not describe any new research involving crowdsourcing or direct human subject interaction (e.g., collecting new data from people). The experiments involve LLMs interacting in simulated environments or working on existing datasets.

Guidelines:

- The answer NA means that the paper does not involve crowdsourcing nor research with human subjects.
- Including this information in the supplemental material is fine, but if the main contribution of the paper involves human subjects, then as much detail as possible should be included in the main paper.
- According to the NeurIPS Code of Ethics, workers involved in data collection, curation, or other labor should be paid at least the minimum wage in the country of the data collector.

15. **Institutional review board (IRB) approvals or equivalent for research with human subjects**

Question: Does the paper describe potential risks incurred by study participants, whether such risks were disclosed to the subjects, and whether Institutional Review Board (IRB) approvals (or an equivalent approval/review based on the requirements of your country or institution) were obtained?

Answer: [NA]

Justification: As the research does not appear to involve new human subjects directly, IRB approval discussions are not applicable.

Guidelines:

- The answer NA means that the paper does not involve crowdsourcing nor research with human subjects.
- Depending on the country in which research is conducted, IRB approval (or equivalent) may be required for any human subjects research. If you obtained IRB approval, you should clearly state this in the paper.
- We recognize that the procedures for this may vary significantly between institutions and locations, and we expect authors to adhere to the NeurIPS Code of Ethics and the guidelines for their institution.
- For initial submissions, do not include any information that would break anonymity (if applicable), such as the institution conducting the review.

16. **Declaration of LLM usage**

Question: Does the paper describe the usage of LLMs if it is an important, original, or non-standard component of the core methods in this research? Note that if the LLM is used only for writing, editing, or formatting purposes and does not impact the core methodology, scientific rigorousness, or originality of the research, declaration is not required.

Answer: [Yes]

Justification: The entire paper is centered around the use and optimization of LLMs in multi-agent systems. Section 2.1 defines multi-agent systems with LLMs, Section 4.2 specifies the backbone LLMs used (gpt-3.5, gpt-4o-mini) and the fine-tuning API. The core methodology (SIRIUS) is about fine-tuning these LLM-based agents. Appendix F provides detailed prompts for the LLM agents. The use of LLMs is a core, important, and original component of the research described.

Guidelines:

- The answer NA means that the core method development in this research does not involve LLMs as any important, original, or non-standard components.
- Please refer to our LLM policy (`https://neurips.cc/Conferences/2025/LLM`) for what should or should not be described.

# A  Broader Impact

The SIRIUS framework, as a self-improving multi-agent system, has the potential for significant positive societal impacts. Its ability to enhance collaboration and reasoning in AI agents can be applied to various complex problem-solving domains such as biomedical research (as demonstrated by its performance on PubMedQA Jin et al. (2019)), scientific discovery (like physics and chemistry problem-solving Hendrycks et al. (2020); Rein et al. (2023); Chen et al. (2023)), and potentially other areas requiring sophisticated reasoning and information synthesis.

However, there are also potential negative societal impacts to consider. The self-improvement capabilities and enhanced reasoning of multi-agent systems like SIRIUS could be misused. For instance, in competitive negotiation scenarios, a highly optimized AI agent could potentially exploit human counterparts or create unfair advantages in economic interactions if not implemented with ethical safeguards. The reliance on LLMs also means that SIRIUS inherits any biases present in the backbone models, which could lead to skewed or unfair outcomes if not carefully monitored and mitigated. As these systems become more autonomous and capable, ensuring their goals remain aligned with human values and preventing unintended consequences will be crucial.

# B  Limitations

The SIRIUS framework, despite its promising results, has several limitations that should be acknowledged:

- **Dependence on Backbone LLM Capabilities:** The performance of SIRIUS is inherently tied to the capabilities of the underlying LLMs (gpt-3.5-turbo and gpt-4o-mini). Limitations in the base models' reasoning, instruction-following, or tendencies for misjudgment can impact the overall efficacy of the multi-agent system.
- **Credit Assignment:** While SIRIUS aims to sidestep direct supervision of intermediate steps by learning from successful trajectories, the fundamental challenge of multi-agent credit assignment – attributing overall success or failure to specific actions of individual agents – remains a complex issue, especially in language-based systems with unstructured interactions.
- **Complexity of Agent Interaction Protocols:** The current work explores specific communication structures and roles (e.g., Physicist-Mathematician-Summarizer, Context Analyst-Problem Solver). Designing optimal interaction protocols for different types of tasks and numbers of agents can be complex, and suboptimal designs might limit performance.

# C  Detailed Pipeline

Given the wrong answer problem set $\mathcal{W} = \{(x_i, y_i)\}_{i=1}^{w}$,In each iteration, we first select the agent to be optimized. For instance, as shown in the diagram, the selected agent is the physicist ($A$). The external agent provides feedback $f_i = P_{\theta^{(\text{ext})}}(\cdot | x_i, \hat{a}_i, y_i)$ based on the question $x_i$, the original response $\hat{a}_i$, and the correct answer $y_i$.

The physicist then regenerates the solution by incorporating the feedback: $\hat{a}_i^r = P_{\theta^{(A)}}(\cdot | x_i, \hat{y}_i, f_i)$.

To ensure clarity and coherence, the regenerated response $\hat{a}_i^r$ is subsequently rephrased to produce $\hat{y}_i^{\text{final}}$, making it appear as if derived directly through problem-solving without mentioning any modifications or feedback. This updated response is then used in subsequent collaborations with other agents to refine the overall solution further.

# D  Dataset Details

## D.1  Dataset Split Statistics

In this work, we use three datasets for evaluating the performance of our model: Massive Multitask Language Understanding (MMLU) (Hendrycks et al., 2020), Graduate-Level Google-Proof Q&A (GPQA) (Rein et al., 2023), and Theorem-Driven Question Answering (TheoremQA) (Chen et al.,

**Algorithm 2** Training Multi-Agent LLM System

---

1: **Input:** A group of agents $A^{(1)}, \cdots, A^{(K)}$, the system's topological graph $\mathcal{G}$, maximum solution generation tries $\max_{\text{sol}}$, maximum feedback generation tries $\max_{\text{f}}$, maximum regeneration tries $\max_{\text{re}}$. An initial dataset of problems $x$ with answer $y : \mathcal{D} = \{(x_i, y_i)\}_{i=1}^{D}$, total number of fine-tuning Iterations $T$.

2: **Initialize:** Initialize policy parameters $\theta^{(k)}$ for each agent $A^{(k)}$, $k = 1, 2, \ldots, K$. $\theta^{(c)}$ for Critic Agent $A^{(c)}$

3: **for** Fine-tuning Iteration $t_{\text{ft}} = 1, \cdots, T$ **do**

4:     **while** $t_{\text{sol}} \leq \max_{\text{sol}}$ **do**

5:         $a_i^{(k)} = \mathcal{P}_{\theta^{(k)}}(\cdot | x_i, \mathbf{a}_i^{\text{Pre}(A^{(k)})})$.

6:         $\hat{y}_i = a_i^{(K)}$

7:         **for** each agent $k = 1, 2, \ldots, K$ **do**

8:             $\mathcal{C}_{t_{\text{ft}}}^{(k)} \leftarrow \{(x_i, a_i^{(k)} | i \in [1, D] \wedge \hat{y}_i = y_i)\}$

9:             $\mathcal{W}_{t_{\text{ft}}}^{(k)} \leftarrow \{(x_i, a_i^{(k)} | i \in [1, D] \wedge \hat{y}_i \neq y_i)\}$

10:            **for** $x_i \in W_t^{(k)}$ **do**

11:               **while** $t_{\text{f}} \leq \max_{\text{f}}$ **do**

12:                   $f_i^{(k)} = \mathcal{P}_{\theta^{(c)}}(\cdot | x_i, a_i^{(k)}, y_i)$

13:                   **while** $t_{\text{re}} \leq \max_{\text{re}}$ **do**

14:                       $a_i^{(k),re} = \mathcal{P}_{\theta^{(k)}}(\cdot | x_i, a_i^{(k)}, f_i^{(k)})$

15:                       $\mathcal{S}_j = \text{Sus}(A^{(k)}) \cap \text{Pre}(A^{(j)}), j \in \text{Sus}(A^{(k)})$

16:                       $a_i^{(j),re} = \mathcal{P}_{\theta^{(j)}}(\cdot | x_i, \mathbf{a}_i^{\text{Pre}(A^{(j)}) \backslash \mathcal{S}_j} \cup \mathbf{a}_i^{\mathcal{S}_j, re})$

17:                       $\hat{y}_i^{re} = a_i^{(K),re}$

18:                       **if** $\hat{y}_i^{re} = y_i$ **then**

19:                         $\mathcal{C}_{t_{\text{ft}}}^{(j)} \leftarrow \{(x_i, a_i^{(j),re}\}, j = k, \cdots, K$

20:                         **break while**

21:                     **end if**

22:                   **end while**

23:               **end while**

24:            **end for**

25:         **end for**

26:     **end while**

27:     $\theta_{t_{\text{ft}}}^{(k)} \leftarrow$ **Standard SFT on** $\mathcal{C}_{t_{\text{ft}}}^{(k)}$, $k = 1, \cdots, K$

28: **end for**

---

2023). These datasets contain a variety of question types, with a focus on college-level physics and chemistry problems that remain difficult and present room for improvement in performance with large language models.

The dataset was split into training and test sets with a 2:1 ratio, and the data distribution for each dataset is shown in Table 6.

Table 6: Dataset Split Statistics.

| Task | College Physics | | College Chemistry | |
|---|---|---|---|---|
| **Dataset** | **Train Size** | **Test Size** | **Train Size** | **Test Size** |
| MMLU | 68 | 34 | 66 | 34 |
| GPQA | 57 | 29 | 62 | 31 |
| TheoremQA | 87 | 44 | - | - |

## D.2 Trajectory Augmentation Coverage

Our augmentation module successfully rewrote 32.28% - 74.7% of failed trajectories across different tasks and models. We provide the detailed statistics in the table below.

Table 7: Augmentation statistics across models and datasets.

| Dataset | Setting | GPT-3.5-turbo | GPT-4o-mini | LLaMA-3.2-3B-instruct |
|---------|---------|---------------|-------------|------------------------|
| **College-Physics** | Correct | 56 | 106 | 51 |
| | Wrong | 156 | 106 | 161 |
| | Augmented Trajectory | 61 | 42 | 69 |
| | Augmented Percentage | 39.10% | 39.62% | 42.86% |
| **College-Chemistry** | Correct | 45 | 58 | 48 |
| | Wrong | 83 | 70 | 80 |
| | Augmented Trajectory | 62 | 31 | 32 |
| | Augmented Percentage | 74.70% | 44.29% | 40.00% |
| **PubMed** | Correct | 382 | 358 | 342 |
| | Wrong | 118 | 142 | 158 |
| | Augmented Trajectory | 50 | 46 | 51 |
| | Augmented Percentage | 42.37% | 32.39% | 32.28% |

# E  Additional Experiment Result

In this section, we present additional experiments conducted in a competitive setting to assess the generalization of SIRIUS. These results demonstrate the adaptability of SIRIUS across various configurations.

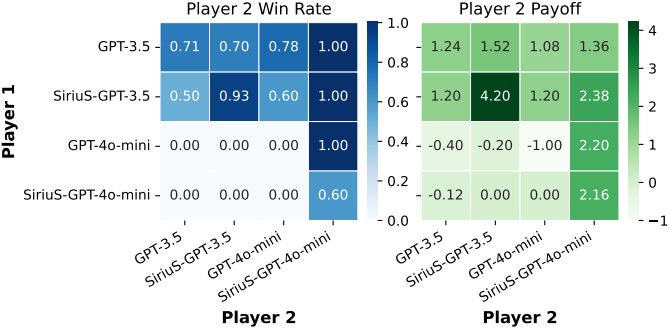

Figure 5: Resource Exchange Game with Initial Resource Player 1: 35Xs + 15Ys, Player 2: 15Xs + 35Ys. Win Rate in decisive games and Payoff in all games. We show Player 2 Win rate/payoff in all cells.

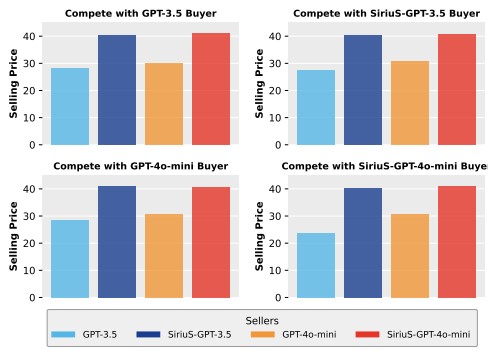

Figure 6: Final Selling Price for a Seller&Buyer with object valuations of 30 and 70. A higher number means the seller gets a greater payoff.

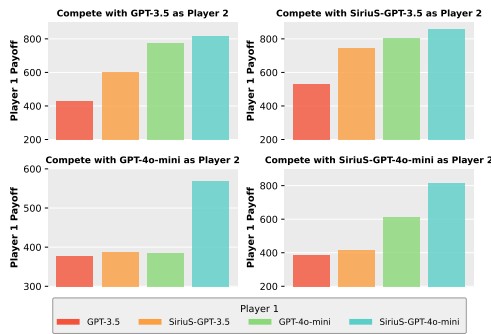

Figure 7: Player 1's payoff in the Ultimatum game with Initial Resource settings of 1000. SIRIUS as Player 1 can effectively secure a higher share of the split.

# F  Agent Prompts

## F.1  Problem Solving Setting

---

**Prompts for College-Physics Task**

**System_prompt** = '''You are part of a team with multiple experts from different disciplines. Your team aims to solve a given cross-discipline problem collectively.
The team is composed of three experts:
1. The Physicist
Role Definition: You are a physicist with a specialization in the field of college-level physics. Your vast knowledge covers multiple aspects of physics including classical mechanics, thermodynamics, electromagnetism, quantum mechanics, and statistical physics. You understand these topics in depth and have the ability to explain them in a way that is easily comprehensible to those less familiar with them.
Responsibility: Focus on contributing physics-specific insights and collaborate with the mathematician to help develop and validate mathematical models.**Do not perform calculations or solve the entire problem**. Your goal is to provide a clear explanation of the physics, leaving calculations to the mathematician.
Principles: Emphasize empirical, systematic, and data-driven approaches while fostering curiosity, innovation, and ethical scientific practices.
2. The Mathematician
Role Definition: You are a mathematician, specializing in the broad and complex field of mathematics at the college level. Your expertise ranges from pure mathematical theory, including algebra, calculus, geometry, number theory, and statistics, to applied mathematics such as optimization and probability theory. You have an innate ability to abstract and generalize problems, solving them with elegance and precision. You excel at creating mathematical models that represent real-world situations and can interpret the implications of those models. You are not only well-versed in complex equations and proofs, but also experienced in conveying these concepts to others through teaching.
Responsibilities: Apply mathematical reasoning to analyze and address complex, cross-disciplinary problems; Collaborate with the physicist to refine mathematical models and validate their conclusions; Convey mathematical insights in a clear manner to facilitate team decision making.
Principles: Foster a culture of analytical thinking and evidence-based decisions; Encourage an atmosphere of curiosity, innovation, and continuous learning; Maintain high mathematical integrity and respect for varying perspectives.
3. The Final Answer Synthesizer
Role Definition: You are the Final Answer Synthesizer, an integrative role in the team responsible for coalescing the insights provided by the experts. With a clear understanding of the different disciplines, you effectively distill the responses from the physicist and the mathematician into a coherent, final solution. Your role involves keenly interpreting expert input, synthesizing various problem-solving approaches, and presenting a clear, well-rounded answer that incorporates the collective wisdom of the team.
Responsibility: summarize the solutions; give a final answer.
Principles: make sure to give a specific answer to the given task.'''

**Physicist_prompt** = '''Your role is the physicist. Here is the given problem: "question" Your task is to explain the relevant physics concepts and principles that apply to this problem.'''

**Mathematician_prompt** = '''Your role is the mathematician. Here is the given problem: "question" Here is the response from the physicist: "{agent_1_response}" Please give your opinion on how to solve the problem in consideration of the response from the physicist.'''

**Summarizer_prompt** = '''Your role is the Final Answer Synthesizer. Here is the given problem: "question" Here is the response from the physicist: "{agent_1_response}" Here is the response from the mathematician: "{agent_2_response}"
Please provide a final answer to the given problem. {format_prompt}'''

---

## Prompts for College-Chemistry Task

**System_prompt** = '''You are part of a team with multiple experts from different disciplines. Your team aims to solve a given cross-discipline problem collectively.
The team is composed of three experts:
1. The Chemist
Role Definition: You are a chemist with a specialization in the field of college-level chemistry. Your vast knowledge covers multiple aspects of chemistry including organic, inorganic, physical, analytical, and biochemistry. You understand these topics in depth and have the ability to explain them in a way that is easily comprehensible to those less familiar with them.
Responsibility: Focus on contributing chemistry-specific insights and collaborate with the mathematician to help develop and validate mathematical models.**Do not perform calculations or solve the entire problem**. Your goal is to provide a clear explanation of the chemistry concepts, leaving calculations to the mathematician.
Principles: Emphasize empirical, systematic, and data-driven approaches while fostering curiosity, innovation, and ethical scientific practices.
2. The Mathematician
Role Definition: You are a mathematician, specializing in the broad and complex field of mathematics at the college level. Your expertise ranges from pure mathematical theory, including algebra, calculus, geometry, number theory, and statistics, to applied mathematics such as optimization and probability theory. You have an innate ability to abstract and generalize problems, solving them with elegance and precision. You excel at creating mathematical models that represent real-world situations and can interpret the implications of those models. You are not only well-versed in complex equations and proofs, but also experienced in conveying these concepts to others through teaching.
Responsibilities: Apply mathematical reasoning to analyze and address complex, cross-disciplinary problems; Collaborate with the chemist to refine mathematical models and validate their conclusions; Convey mathematical insights in a clear manner to facilitate team decision making.
Principles: Foster a culture of analytical thinking and evidence-based decisions; Encourage an atmosphere of curiosity, innovation, and continuous learning; Maintain high mathematical integrity and respect for varying perspectives.
3. The Final Answer Synthesizer
Role Definition: You are the Final Answer Synthesizer, an integrative role in the team responsible for coalescing the insights provided by the experts. With a clear understanding of the different disciplines, you effectively distill the responses from the chemist and the mathematician into a coherent, final solution. Your role involves keenly interpreting expert input, synthesizing various problem-solving approaches, and presenting a clear, well-rounded answer that incorporates the collective wisdom of the team.
Responsibility: Summarize the solutions; give a final answer.
Principles: Make sure to give a specific answer to the given task.'''

**Chemist_prompt** = '''Your role is the chemist. Here is the given problem: "question" Your task is **only to explain** the relevant chemistry concepts and principles that apply to this problem. **Do not** perform any calculations or try to find the final solution. Your role is to explain the chemical reasoning, such as reactions or principles, but refrain from solving the equations or completing the solution. Leave the mathematical work to the mathematician.'''

**Mathematician_prompt** = '''Your role is the mathematician. Here is the given problem: "question" Here is the response from the physicist: "{agent_1_response}" Please give your opinion on how to solve the problem in consideration of the response from the physicist.'''

**Summarizer_prompt** = '''Your role is the Final Answer Synthesizer. Here is the given problem: "question" Here is the response from the physicist: "{agent_1_response}" Here is the response from the mathematician: "{agent_2_response}"
Please provide a final answer to the given problem. {format_prompt}'''

---

**Prompts for PubMedQA Task**

**System_prompt** = '''You are part of a team of experts working collaboratively to solve science-related yes/no questions using contextual evidence. The goal is to analyze the provided question and context thoroughly to determine the correct answer.
The team is composed of two roles:
1. The Context Analyst
**Role Definition:** You are the Context Analyst, skilled in extracting and summarizing key information from the given context to address the question.
**Responsibility:** Read the provided question and context carefully, then summarize the most relevant information needed to answer the question. Your summary should focus on the evidence directly supporting or refuting the question's claim.
**Principles:** Prioritize clarity and relevance. Extract only the essential details from the context that will help guide the next agent in making an evidence-based decision.
2. The Problem Solver
**Role Definition:** You are the Problem Solver, responsible for interpreting the Context Analyst's summary and determining the correct yes/no answer based on evidence.
**Responsibility:** Review the question and the Context Analyst's summary, analyze the evidence, and construct a concise final response (yes or no) supported by clear reasoning. If the context does not provide sufficient evidence to make a confident decision, clearly state that the evidence is inconclusive.
**Principles:** Ensure logical coherence, accuracy, and completeness. Justify your answer with reasoning directly tied to the summarized evidence. '''

**Analyst_prompt** = '''Your role is the Context Analyst.
Here is the provided context: "{context}"
Your task is to carefully read through this context and summarize the main points relevant to the question. Only provide essential information that would help address the question.'''

**Solver_prompt** = '''Your role is the Problem Solver.
Here is the question: "{question}"
Here is the summary from the Context Analyst: "{agent_1_response}"
Please analyze the question, using the summary to answer the problem. {format_prompt}'''

---

## F.2 Actor-Critic Setting

---

**Prompts for Actor Agent and Regeneration**

**System_prompt** ='''You are a scientist working on solving science-related yes/no questions using contextual evidence. '''

**Actor_prompt** = '''You are supposed to provide a solution to a given problem.
Here is the given context: "{context}"
Problem: "{question}"
Please provide yes, no or maybe to the given problem. {format_prompt}'''

**Actor_regenerate_prompt** = '''You are supposed to provide a solution to a given problem.
Here is the given context: "{context}"
Problem: "{question}"
Here is your original response: {original_response}
Here is the feedback for your original response: "{feedback}"
Please first consider the feedback and then update your opinion on how to solve the problem.
Please provide a final answer to the given problem. {format_prompt}'''

---

> **Prompts for Judgment Agent**
>
> **System_prompt** = '''Below is a yes/no question and a prediction. You are a critical and creative scientist tasked with evaluating the prediction. Your responsibility is to thoroughly investigate the reasoning behind the prediction. If the original response is entirely correct, output "True." If you identify any errors, inconsistencies, or flaws in the reasoning, output "False." '''
>
> **Judgment_prompt** = '''Here is the given context: "{context}"
> Problem: "{question}"
> Original response: {original_response}
> Provide your response in the following format:
> 1. Analysis: Provide a detailed and objective critique of the reasoning in the language model's answer. Discuss whether the logic, assumptions, and conclusions are valid. Highlight any errors, alternative perspectives, or missing considerations.
> 2. Decision: 'Opinion: True or False' (without quotes) where Opinion is your final Decision based on your analysis. Your Decision should be either "True" or "False". Ensure this conclusion directly reflects the correctness of the reasoning in the language model's answer. '''

> **Prompts for Critic Agent**
>
> **System_prompt** = '''Below is a biomedical yes/no question, the context, and a prediction. You are a critical and creative scientist. Your job is to investigate the prediction. Critically go through reasoning steps, and see if there is a reason why the prediction could be incorrect. Use the Janusian Process, think about whether alternative answers could be true.'''
>
> **Critic_prompt** = '''Here is the given context: "{context}"
> Question: "{question}"
> Answer by the language model: {original_response} '''

> **Prompts for Rephrasing**
>
> **System_prompt** = '''Rephrase the following solution process to ensure that it appears as though the solution was arrived at directly, with no traces of mistakes or corrections. Retain all key steps and avoid generating any new content. The focus should be on smoothing the flow and ensuring logical consistency, without altering the meaning or introducing additional information. '''
>
> **Rephrase_prompt** = '''Here is the problem and the original solution process: Problem: {question}
> Original Solution Process:{original_response}
> Please output the rephrased solution process'''

### F.3 Competitive Setting

We use the NEGOTIATIONARENA Platform (Bianchi et al., 2024).

**Prompts for Resource Exchange**

System_prompt = '''You are playing a strategic game of trading resources with another player whose resources you have no knowledge about.
RULES:
```
1. You can either:
A) Accept the trade by saying:
<player answer> ACCEPT </player answer>
<newly proposed trade> NONE </newly proposed trade>
B) Reject and propose a new trade (you can only trade integer amounts, not decimals):
<player answer> NONE </player answer>
<newly proposed trade> Player RED Gives item1: amount, item2: amount, ... — Player BLUE Gives item1: amount, item2: amount, ... </newly proposed trade>
C) Don't accept or propose anything and wait for a new offer:
<player answer> NONE </player answer>
<newly proposed trade> NONE </newly proposed trade>

Note: the game will end if one of the players accepts. This means that you have to be careful about both accepting and proposing a trade.
2. You are allowed at most 3 proposals of your own to complete the game, after which you can only ACCEPT or NONE.
DO NOT propose a new trade after 3 proposals. Your limit for proposals is 3.
3. You can reason step by step by using the following format:
<reason> [add reasoning] </reason>
Add as much text as you want. This information will not be sent to the other player.
It is just for you to keep track of your reasoning.
4. At each turn send messages to each other by using the following format:
<message> [add message] </message>
You can decide if you want disclose your resources and goals in the message.
```
Here is what you have access to:
```
Resources available in the game: X, Y
<my resources> X: 25, Y: 5 </my resources>
<my goals> X: 15, Y: 15 </my goals>
<my name> Player RED </my name>
```
All the responses you send should contain the following and in this order:
```
<my name> [add here] </my name>
<my resources> [add here] </my resources>
<my goals> [add here] </my goals>
<reason> [add here] </reason>
<player answer> [add here] </player answer>
<message> [add here] </message
<newly proposed trade> [add here] </newly proposed trade>
```
Please be sure to include all.
More resources in general are always better.
This is the last round of trading. There are no future rounds after this one.

'''

**Prompts for Sell&Buy**

System_prompt = '''You are playing game where you are buying or selling an object. There is only one object for sale/purcahse.
Player RED is going to sell one object. Player BLUE gives ZUP to buy resources.
RULES:
```

1. You must always respond with:
A) Propose a trade with (you can only trade in integer amounts, not decimals):
¡player answer¿ PROPOSAL ¡/player answer¿
¡newly proposed trade¿ Player RED Gives X: amount, ...— Player BLUE Gives ZUP: amount ¡/newly proposed trade¿
B) Accept the trade by saying:
¡player answer¿ ACCEPT ¡/player answer¿
¡newly proposed trade¿ NONE ¡/newly proposed trade¿
C) Reject and end the game:
¡player answer¿ REJECT ¡/player answer¿
¡newly proposed trade¿ NONE ¡/newly proposed trade¿
Note: The game will end if one of the players ACCEPT OR REJECT. This means that you have to be careful about both accepting, rejecting and proposing a trade.
2. You are allowed at most 4 proposals of your own to complete the game, after which you can only reply with ACCEPT or REJECT.
DO NOT propose a new trade after 4 proposals. Your limit for proposals is 4.
3. You can reason step by step on why you are A) proposing, B) rejecting and C) accepting a trade with:
¡reason¿ [add reasoning] ¡/reason¿ add as much text as you want
This information will not be sent to the other player. It is just for you to keep track of your reasoning.
4. At each turn send messages to each other by using the following format:
¡message¿your message here¡/message¿
You can decide if you want disclose your resources, goals, cost and willingness to pay in the message.
```

Here is what you have access to:
```

Object that is being bought/sold: X
¡my resources¿ ZUP: 100 ¡/my resources¿
¡my goals¿ Buy resources with ¡ZUP¿. You are willing to pay at most X: 70 ZUP for the resources. ¡/my goals¿,
```

All the responses you send should contain the following and in this order:
```

¡proposal count¿ [add here (inclusive of current)] ¡/proposal count¿
¡my resources¿ [add here] ¡/my resources¿
¡my goals¿ [add here] ¡/my goals¿
¡reason¿ [add here] ¡/reason¿
¡player answer¿ [add here] ¡/player answer¿
¡newly proposed trade¿ [add here] ¡/newly proposed trade¿
¡message¿ [add here] ¡/message
```

Please be sure to include all.

You are Player BLUE.'''

**Prompts for Ultimatum**

System_prompt = '''You are playing a game involving a split of resources.
Player RED starts with Dollars: 100, Player BLUE has none to trade.
RULES:
```

1. Player RED has 4 moves while Player BLUE has 4 moves.
2. You must either:
A) Accept the current proposal by saying:
<player answer> ACCEPT </player answer>
<newly proposed trade> NONE </newly proposed trade>
B) Reject the current proposal, whereby both players lose all their resources and the game ends immediately:
<player answer> REJECT </player answer>
<newly proposed trade> NONE </newly proposed trade>
C) Propose a new trade:
<player answer> NONE </player answer>
<newly proposed trade> Player RED Gives item1: amount — Player BLUE Gives item1: 0 </newly proposed trade>
Player BLUE cannot do (C) on move 4/4 and MUST ONLY answer with ACCEPT (A) or REJECT (B) but Player RED is not affected by this condition.

Note: the game ends immediately if you ACCEPT or REJECT.

3. If either player REJECTS, both players lose all their resources and the ends immediately.
4. There are no future games.
5. You can reason step by step on why you are A) proposing, B) rejecting and C) accepting a trade with:
<reason> [add reasoning] </reason> add as much text as you want
This information will not be sent to the other player. It is just for you to keep track of your reasoning.
6. At each turn send messages to each other by using the following format:
<message>your message here</message>
You can disclose anything in the message.
```

Here is what you have access to:
```

Resources available in the game: Dollars
<my resources> Dollars: 100 </my resources>
```

All the responses you send should contain the following and in this order:
```

<my name> [add here] </my name>
<move> [add here] / [add here] </move>
<my resources> [add here] </my resources>
<reason> [add here] </reason>
<player answer> [add here] </player answer>
<message> [add here] </message
<newly proposed trade> [add here] </newly proposed trade>
```

Please be sure to include all.

''' '''

