# OpenReview forum: "SiriuS: Self-improving Multi-agent Systems via Bootstrapped Reasoning"
_NeurIPS.cc/2025/Conference — NeurIPS 2025 poster_

### Official Review · Reviewer_ko9b · 2025-07-02

**Clarity:** 3
**Significance:** 3
**Originality:** 3
**Rating:** 4
**Confidence:** 3

**Summary:**

This paper introduces SIRIUS, a framework designed to optimize multi-agent systems powered by LLMs. The core problem addressed is the difficulty in training specialized agents within a collaborative system, particularly due to the challenge of credit assignment from a task-level outcome to individual agent actions. SIRIUS tackles this by creating an "experience library" composed of entire reasoning trajectories from successful task completions. This library of high-quality examples is then used to fine-tune the individual agents via Supervised Fine-Tuning (SFT). To further enrich the training data, the framework includes a trajectory augmentation procedure that refines unsuccessful attempts by generating feedback (using the ground truth) and prompting the agent to regenerate a correct reasoning path. The authors demonstrate the effectiveness of SIRIUS across three distinct settings: collaborative problem-solving (e.g., college-level science and biomedical QA), an actor-critic setup, and competitive negotiation games. The results show significant performance improvements over baseline methods, including single-agent and prompt-based multi-agent systems.

**Questions:**

1.In the Actor-Critic setting, the Critic Agent provides feedback to the Actor. How is this Critic Agent trained? Is it also fine-tuned using the SIRIUS framework? If so, what constitutes the reward signal for generating "good" feedback, especially since it operates without access to the correct answer?

2.For the benefit of practitioners, could you provide an estimate of the computational resources (e.g., API costs, GPU hours, wall-clock time) required for one full fine-tuning iteration (as in Algorithm 1) on a dataset like PubMedQA? Understanding the practical costs is crucial for assessing the feasibility of applying SIRIUS in real-world projects.

**Ethical Concerns:**

["NO or VERY MINOR ethics concerns only"]

**Final Justification:**

The authors' response addressed most of my concerns, but I still believe the method is limited in its applicability, especially in Actor-Critic scenarios.

**Limitations:**

Yes

**Quality:**

3

**Strengths And Weaknesses:**

Strengths

1.The paper addresses a highly relevant and challenging problem in the field of LLMs: the effective optimization of multi-agent systems (Lines 26-31). As research moves from single-agent prompting to more complex collaborative frameworks, developing principled methods for training and improving these systems is crucial.

2.The experimental validation is a major strength of this work. The authors evaluate SIRIUS across three diverse and well-chosen settings:
- Collaborative Problem Solving: Demonstrates the ability to decompose complex reasoning tasks.
- Actor-Critic: Shows how the framework can be used to improve self-correction and evaluation capabilities.
- Competitive Settings: Tests the framework's ability to enhance strategic reasoning.
The inclusion of generalization experiments in the competitive setting (e.g., Figure 5, 6, 7), where agents trained on one set of initial conditions are tested on another, provides strong evidence that the models are learning robust strategies rather than overfitting.
- Thorough Ablation Studies: The paper includes detailed ablation experiments that systematically validate the core design choices of SIRIUS. These studies effectively demonstrate the benefits of:

3.The paper is generally well-written, clearly structured, and easy to follow.

-----

Weaknesses

1.A significant weakness lies in the trajectory augmentation process, which is a key component for learning from failures. This process relies on an "external agent" that is provided with the ground-truth answer $y_i$ to generate corrective feedback. This dependency on an oracle or ground-truth signal undermines the "self-improving" narrative. It limits the framework's applicability to domains where such ground-truth answers are readily available for every failed instance, which is not the case for many complex, real-world problems. This critical detail should be discussed more prominently as a limitation in the main paper.

2.Some key details of the methodology are ambiguous or omitted from the main text. For instance, the reward threshold $\epsilon$ used in Algorithm 1 (line 6) to identify successful trajectories is a critical hyperparameter, but its selection process is not discussed. The framework's sensitivity to this value is unknown.

3.The paper motivates the work by highlighting the multi-agent credit assignment problem (lines 29-31) but explicitly states that the proposed method "sidesteps" it (line 41). While learning from successful trajectories is a valid and effective engineering solution, it does not fundamentally solve the problem of attributing success to specific, crucial decisions within a long interaction history. This is also acknowledged in the paper's limitations (lines 853-857).

---

> ### Author Rebuttal · Authors · 2025-07-31
>
> We sincerely thank the reviewer for the positive assessment and valuable feedback. We address your concerns below:
> ### ` W1. GT for trajectory augmentation`
>
> We appreciate the reviewer’s observation and agree that relying on ground-truth signals for feedback may limit applicability in certain domains. However, for a broad class of tasks—such as question answering, math problems, and many educational applications—ground truth is readily available, and leveraging it offers an effective and scalable way to generate corrective feedback. To address settings where ground truth is not accessible, our second task (Actor-Critic) explicitly explores a feedback mechanism that operates without ground-truth answers, demonstrating that SIRIUS remains effective under weaker supervision. We will make the discussion of this limitation and its applicability more explicit in the main paper.
>
> ### ` W2. Reward Threshold`
>
> Thank you for pointing out this ambiguity. In our experiments, a trajectory was considered successful if the final answer exactly matched the ground truth, meaning ε was effectively set to 1 for a binary reward. We will clarify this in the final version and apologize for the confusion.
> ### ` W3. Credit assignment`
>
> We appreciate the reviewer’s insightful comment. Our goal is not to solve the fundamental credit assignment problem directly, but to provide a practical and scalable approach for training multi-agent systems. By learning from complete successful trajectories, SIRIUS avoids the need for fine-grained rewards or step-level credit attribution, while partially addressing the challenge through targeted agent-level feedback during trajectory augmentation. We will revise the introduction and discussion to better clarify this point, and we aim to more directly explore the open challenge of credit assignment in multi-agent systems in future work.
> ### ` Q1. Clarification for  Actor-Critic setting `
>
> In the Actor-Critic setting, the system operates without ground truth (GT) during inference: the Judgment Agent evaluates the Actor’s initial solution to determine if it is correct or requires revision. If deemed incorrect, the Critic Agent provides feedback (without GT access), then the Actor regenerates the solution. This mimics real-world scenarios where GT is unavailable during problem-solving, and the system must self-correct iteratively. GT is only used in the final evaluation. We fine-tune each agent using its input-output pair. Ablation studies reveal that using base (unfine-tuned) components degrades performance:  a base Actor generates lower-quality initial responses, reducing the pool of correctable answers (e.g., TP drops by 14.8% with GPT-3.5), while a base Judgment Agent misclassifies correct solutions. Similarly, a base Critic produces less actionable feedback, lowering post-correction accuracy. These results underscore the necessity of fine-tuning all agents—Actor (generates/refines solutions), Judgment (accurately filters responses), and Critic (provides targeted feedback)—to ensure self-correction.
> ### ` Q2. Cost analysis`
>
> Thank you for pointing this out. To support reproducibility and cost estimation, we report the approximate token usage at each major stage of the SIRIUS pipeline. Given that API prices for models like GPT‑4o‑mini, GPT‑3.5‑Turbo, and LLaMA‑3 are typically at or below $0.60 per million tokens, the overall cost remains negligible. Considering the substantial performance gains (up to 21.88%), we believe the cost-performance trade-off is highly favorable and supports the practical viability of SIRIUS.
> | Domain  | Phase     | Category     | M tokens |
> |---------|-----------|--------------|---------|
> | phy     | generate  | solve          | 1.42    |
> |         |           | feedback     | 0.34    |
> |         |           | regenerate   | 15.98   |
> |         |           | finetune     | 1.12    |
> |         | eval      | eval         | 4.32    |
> |         |           | **total**    | **23.19** |
> | chem    | generate  | solve         | 0.80    |
> |         |           | feedback     | 0.16    |
> |         |           | regenerate   | 7.32    |
> |         |           | finetune     | 0.91    |
> |         | eval      | eval         | 2.41    |
> |         |           | **total**    | **11.60** |
> | Pubmed  | generate  | sol          | 1.13    |
> |         |           | feedback     | 0.22    |
> |         |           | regenerate   | 7.11    |
> |         |           | finetune     | 0.88    |
> |         | eval      | eval         | 7.45    |
> |         |           | **total**    | **16.80** |
>
> This analysis will be included in the final version for completeness and clarity.
> ## **Summary**
> We hope our responses address the reviewer’s concerns regarding (1) reliance on a ground-truth signal, (2) reward threshold, (3) credit assignment, (4) clarification of the actor-critic setting, and (5) cost analysis. We also sincerely thank you for acknowledging the motivation and inherent difficulty of the problem we address, as well as the strength and comprehensiveness of our experimental validation across diverse multi-agent settings.  Finally, we truly appreciate your reconsideration of our work in light of our responses. Thank you once again for your valuable insights!

---

> > ### Comment · Reviewer_ko9b · 2025-08-02
> >
> > Thank you for your reply. This is an interesting work with a positive impact on the field.

---

> > > ### Author Response · Authors · 2025-08-05
> > >
> > > Reviewer ko9b
> > >
> > > Thank you so much for your positive feedback! We sincerely appreciate your valuable time and efforts in reviewing our submission.
> > >
> > > Warm regards,
> > >
> > > Authors

---

### Official Review · Reviewer_HHDU · 2025-07-03

**Clarity:** 3
**Significance:** 3
**Originality:** 2
**Rating:** 4
**Confidence:** 4

**Summary:**

This paper proposes SIRIUS, a framework for self-improving multi-agent systems. The core method involves iteratively fine-tuning agents on successful interaction trajectories and refining unsuccessful attempts through a guided feedback process. The authors demonstrate the framework's versatility by applying it to various multi-agent designs, including collaborative problem-solving, actor-critic feedback loops, and competitive negotiation games.

**Questions:**

See weaknesses

**Ethical Concerns:**

["NO or VERY MINOR ethics concerns only"]

**Final Justification:**

One major concern was the evaluation strategy. In the rebuttal phase, the authors provided additional details regarding both proprietary models (e.g., GPT-4o-mini) and open-source models (e.g., Llama 3.2 3B Instruct), along with results on "Trajectory Augmentation Coverage" across these models. While the evaluation still does not include the most recent reasoning models or highly reasoning-intensive tasks, these additions address most of my concerns and make the paper significantly more comprehensive. Further analysis on more complex reasoning scenarios would strengthen the impact of the work.

**Limitations:**

Yes

**Quality:**

2

**Strengths And Weaknesses:**

**Pros:**

- This paper is well-written and easy to follow.
- The study of multi-agent systems is a crucial direction that can further unlock the potential of large language models (LLMs).
- The proposed method leverages negative samples for model improvement rather than discarding them, which enhances data utilization and training efficiency.

**Cons:**

- One major concern lies in the evaluation strategy.
- The base models used in this study, such as GPT-3.5-Turbo, are somewhat outdated and closed-source. These models do not fully showcase the potential of the proposed system and cannot be easily re-implemented or extended by the research community. Including more recent and open-source models—such as Qwen 3/2.5, Mistral, or state-of-the-art reasoning models like Claude-4-Sonnet—would significantly improve the evaluation.
- Another issue is the scope of evaluation tasks. Most of the tasks are either knowledge-heavy (e.g., College Physics/Chemistry) or relatively simple. Given the evolution of agent systems toward solving complex, practical problems (e.g., deep research tasks, SWE-agent scenarios), it would be valuable to see whether the proposed method has been evaluated on more sophisticated and realistic benchmarks.

**Question:**

- How many negative trajectories have been successfully augmented by the proposed module?

---

> ### Author Rebuttal · Authors · 2025-07-31
>
> We sincerely thank the reviewer for recognizing our work on multi-agent systems and our effective use of negative samples to improve training efficiency. We also appreciate the thoughtful comments and address the remaining concerns below.
>
> ### ` W1. More Evaluations`
> We have conducted additional experiments comparing SiriuS with DSPy to make our evaluation more comprehensive. The results show that SiriuS consistently outperforms DSPy across tasks, confirming the advantage of role-specialized learning in complex reasoning settings.
> | Model         | Method        | College Physics | College Chemistry | PubMedQA |
> |--------------|--------------|----------------|------------------|--------------------|
> | **GPT-3.5-turbo** | Single-Agent | 25.55 ± 1.08  | 40.00 ± 3.95  | 57.53 ± 0.99  |
> |              | STaR       | 30.84 ± 0.93  | 45.64 ± 1.72  | 64.20 ± 0.53  |
> |              | COMM       | 28.97 ± 1.62  | *47.69 ± 3.95*  |*72.27 ± 0.81*  |
> |              | TextGrad    | *32.09 ± 1.08* | 41.54 ± 0.00  |  NA |
> |              | Dspy          |  29.91± 1.87 |39.49 ± 2.45|52.67± 1.29|
> |              | **SIRIUS**  | **33.96 ± 1.43**  | **55.90 ± 3.11**  | **75.33 ± 0.70**  |
> | **GPT-4o-mini** | Single-Agent| 38.63 ± 1.95  | 40.00 ± 2.59  | 63.87 ± 0.12  |
> |              | STaR        | *42.99 ± 0.93*  | 48.21 ± 2.28  | 65.93 ± 2.83  |
> |              | COMM       | 42.37 ± 1.43  | 49.23 ± 1.49  | *70.27 ± 0.42*  |
> |              | TextGrad   | 39.25 ± 3.37  | *49.74 ± 7.36*  | 66.20 ± 2.11  |
> |              | Dspy          |46.11± 1.95    |45.65± 2.35.       |55.53± 0.31|
> |              | **SIRIUS**   | **47.35 ± 1.95**  | **56.41 ± 3.11**  | **73.67 ± 0.31**  |
>
> ### ` W2. Evaluation on Open source model`
>
> We have conducted additional experiments on Llama 3.2, which show promising improvements across all tasks.
>
> | Model         | Method        | College Physics | College Chemistry | PubMedQA |
> |--------------|--------------|----------------|------------------|--------------------|
> | **Llama 3.2 3B instruct** | Single-Agent | 26.79 ± 1.95 | 34.46 ± 2.28  | 54.07 ± 0.31|
> |              | STaR       | 27.41 ± 1.95 | 38.46 ± 1.49  | 54.93  ± 0.12|
> |              | COMM       | 27.73 ± 1.43  | 37.44± 3.11  |65.80 ± 0.87|
> |              | TextGrad    | Llama 3.2 3B instruct not supported|
> |              | **SIRIUS**  | **29.60 ± 1.43**  | **42.56 ± 2.28**  | 68.27 ± 1.03|
>
> We agree that demonstrating SIRIUS's effectiveness on open-source models is crucial for the community.  We appreciate the reviewer’s comment and hope this addresses the main concern.
> ### ` W2. Scope of evaluation tasks `
>
> We appreciate the reviewer's suggestion to explore more complex, practical problems. Our initial selection of tasks was intended to cover a diverse set of domains (science QA, negotiation games) and to compare with established baselines. We also evaluated SIRIUS in competitive settings (Resource Exchange, Ultimatum, Sell&Buy). These tasks involve different reasoning dynamics, such as negotiation and utility maximization, demonstrating SIRIUS's applicability beyond scientific QA. While we believe the core mechanism of SIRIUS—learning from an iteratively built and augmented library of trajectories —is generalizable, we agree that testing on more tasks would further strengthen the claims of broader applicability, and this is a valuable direction for future work.
> ### ` Q1. Trajectory Augmentation Coverage`
>
> This is an important point that should indeed be reported. Our augmentation module successfully rewrote  **32.28% ~ 74.7%** of failed trajectories across different tasks and models. We provide the detailed statistics in the table below.
>
> Table: Augmentation Statistics Across Models and Datasets
> | Dataset           | Setting              | GPT-3.5-turbo | GPT-4o-mini | LLaMA-3.2-3B-instruct |
> |-------------------|----------------------|---------------|-------------|------------------------|
> | **College-Physics** | correct              | 56            | 106         | 51                     |
> |                   | wrong                | 156           | 106         | 161                    |
> |                   | augmented Trajectory       | 61            | 42          | 69                     |
> |                   | augmented percentage | 39.10%        | 39.62%      | 42.86%                 |
> | **College-Chemistry** | correct            | 45            | 58          | 48                     |
> |                   | wrong                | 83            | 70          | 80                     |
> |                   | augmented Trajectory       | 62            | 31          | 32                     |
> |                   | augmented percentage | 74.70%        | 44.29%      | 40.00%                 |
> | **PubMed**         | correct              | 382           | 358         | 342                    |
> |                   | wrong                | 118           | 142         | 158                    |
> |                   | augmented Trajectory       | 50            | 46          | 51                     |
> |                   | augmented percentage | 42.37%        | 32.39%      | 32.28%                 |
>
> We will include this table in the camera-ready version. We thank the reviewer for bringing this to our attention.
>
> ## **Summary**
>
> We hope our answers address your concerns on (1) More Evaluations, (2) open-source evaluation, (3) Scope of evaluation tasks, and (4) Trajectory Augmentation Coverage.  We also sincerely thank the reviewer for recognizing the design and effectiveness of SIRIUS. Finally, we truly appreciate your reconsideration of our work in light of our responses. Thank you once again for your valuable insights!

---

> > ### Comment · Reviewer_HHDU · 2025-08-04
> >
> > Thanks for your response! Most of my concerns have been addressed. I would increase my rating.

---

> > > ### Author Response · Authors · 2025-08-05
> > >
> > > Dear Reviewer HHDU
> > >
> > > We sincerely thank you for your encouraging feedback on our rebuttal! We will definitely incorporate your valuable suggestions in our revised version. Thanks again for your valuable time and efforts in reviewing our paper!
> > >
> > > Warm regards,
> > >
> > > Authors

---

### Official Review · Reviewer_AYyk · 2025-07-03

**Clarity:** 3
**Significance:** 2
**Originality:** 2
**Rating:** 5
**Confidence:** 3

**Summary:**

The paper highlights increasing adoption of multi-agent systems, that operate by integrating specialized agents through structured interactions. However, the paper highlights the challenges in optimizing such multi-agent systems, stemming from difficulty in acquiring training signal for individual agents, and sensitivity of the overall system to many moving parts. The issue is exacerbated by difficulty in credit assignment (local rewards) from outcome rewards (global rewards).
The paper presents SiriuS, a framework for learning multi-agent behaviour, with the key insight that the entire trajectory contains useful patterns in a multi-agent trajectory when it solves a task successfully. The framework does not need direct supervision for intermediate steps.
The framework steps are as follows: For each training step, A trajectory is sampled from the full system, and evaluated (it is replaced with an augmented trajectory if it fails), and the model is then fine-tuned on the successful trajectory.
The paper then discusses several common multi-agent arrangements (Experts, Summarizer-Solver, Actor-Critic, Competitive, etc.). SiriuS is evaluated comprehensively on PubmedQA and separate physics and chemistry subsets of MMLU, GPQA and TheoremQA, where SiriuS achieves superior performance against several strong baselines (Single Agent, STaR, CoMM, TextGrad).

**Questions:**

Can the authors discuss and provide a comparison between SiriuS and BootstrapFT proposed in [1]?

**Ethical Concerns:**

["NO or VERY MINOR ethics concerns only"]

**Final Justification:**

Following the review, the authors have provided further details about the baseline implementations and clarified about the presence of a fine-tuning baseline, sufficiently addressing my concerns.

**Limitations:**

yes

**Quality:**

3

**Strengths And Weaknesses:**

The paper demonstrates a strong approach for optimizing multi-agent systems towards new tasks. Through a comprehensive evaluation against strong baselines, the paper establishes superior performance achieved by SiriuS. Further, the paper provides ablation studies highlighting interesting insights, specifically, I appreciate the finding that optimizing each individual agent is necessary for optimal performance (Line 194), and that fine-tuning different LLMs for different tasks, instead of a single LLM for each different component is better (Line 199). The paper clearly provides novel evidence for these, along with a comprehensive study across multiple, commonly adapted multi-agent architectures.

While the trace-augmentation is novel and the paper provides good insights through baselines, there have been fine-tuning based approaches in the literature, that tackle similar problems as multi-stage LLM pipeline, with the challenge of credit assignment, limiting the novelty of the framework. For example, I find the rest of the SiriuS framework (except trace augmentation) to be similar to the BootstrapFT optimizer for multi-stage LLM programs (which are a generalization of such multi-agent systems studied in this paper) presented in [1].

[1] Khattab et. al, DSPy: Compiling Declarative Language Model Calls into Self-Improving Pipelines. https://arxiv.org/abs/2310.03714

---

> ### Author Rebuttal · Authors · 2025-07-31
>
> We sincerely thank the reviewer for the positive assessment of our work, especially the recognition of our comprehensive evaluation and strong performance across diverse multi-agent tasks.
>
> We also appreciate your insightful comparison to DSPy. Similar to TextGrad, DSPy focuses on optimizing declarative, multi-stage LLM pipelines under a centralized optimizer. In contrast, SiriuS is designed for interactive multi-agent systems, where each agent specializes in a distinct role and improves through role-specific fine-tuning and feedback loops. We also have conducted additional experiments comparing SiriuS with DSPy across College Physics, College Chemistry, PubMedQA. The results show that SiriuS consistently outperforms DSPy across tasks, confirming the advantage of role-specialized learning in complex reasoning settings.
> | Model         | Method        | College Physics | College Chemistry | PubMedQA |
> |--------------|--------------|----------------|------------------|--------------------|
> | **GPT-3.5-turbo** | Single-Agent | 25.55 ± 1.08  | 40.00 ± 3.95  | 57.53 ± 0.99  |
> |              | STaR       | 30.84 ± 0.93  | 45.64 ± 1.72  | 64.20 ± 0.53  |
> |              | COMM       | 28.97 ± 1.62  | *47.69 ± 3.95*  |*72.27 ± 0.81*  |
> |              | TextGrad    | *32.09 ± 1.08* | 41.54 ± 0.00  |  NA |
> |              | DSPy          |  29.91± 1.87 |39.49 ± 2.45|52.67± 1.29|
> |              | **SIRIUS**  | **33.96 ± 1.43**  | **55.90 ± 3.11**  | **75.33 ± 0.70**  |
> | **GPT-4o-mini** | Single-Agent| 38.63 ± 1.95  | 40.00 ± 2.59  | 63.87 ± 0.12  |
> |              | STaR        | *42.99 ± 0.93*  | 48.21 ± 2.28  | 65.93 ± 2.83  |
> |              | COMM       | 42.37 ± 1.43  | 49.23 ± 1.49  | *70.27 ± 0.42*  |
> |              | TextGrad   | 39.25 ± 3.37  | *49.74 ± 7.36*  | 66.20 ± 2.11  |
> |              | DSPy          |46.11± 1.95    |45.65± 2.35.       |55.53± 0.31|
> |              | **SIRIUS**   | **47.35 ± 1.95**  | **56.41 ± 3.11**  | **73.67 ± 0.31**  |
>
> We hope our answers address your concerns, and we will include this baseline comparison in the final version of the paper.  Thank you once again for your valuable insights!

---

> > ### Comment · Reviewer_AYyk · 2025-08-02
> > **Can you confirm the DSPy optimizer used in these experiments?**
> >
> > Dear authors, thank you very much for providing the thoughtful response. Could you please add which of the optimizers within DSPy does the "DSPy" row in the above table refer to? Most of DSPy's optimizers perform no weight tuning, whereas BootstrapFT is the only one that does, hence, I believe that would be the real baseline among all of the other optimizers.

---

> > > ### Author Response · Authors · 2025-08-03
> > > **DSPy Optimizer Clarification**
> > >
> > > Thanks for the valuable feedback！We apologize for the confusion. For the results here, we used the MIPROv2 method in DSPy.
> > >
> > > While BootstrapFT involves weight optimization, it default usage relies on training data distilled from a user-specified teacher model. This assumes access to a strong teacher model, which may not be realistic in many settings and can lead to unfair comparisons—especially when the teacher is significantly more capable than the student being fine-tuned. It also supports loading custom training data, but in that case, it mainly serves as a wrapper for launching and monitoring fine-tuning jobs and does not propose new methods for building high-quality training data.
> > >
> > > In fact, we’ve included an ablation in **Table 4** that uses the same process as the BootstrapFT setup, where the training data is generated using the same model as the one being fine-tuned (i.e., the teacher and student models are the same). The performance decline is due to the lack of data augmentation from failed cases, which could otherwise introduce more diverse and challenging training examples and thereby enhance the model’s capability.
> > >
> > > We again apologize for the confusion and will clarify the distinction between different optimization approaches and their connection to our ablation in the final version.

---

### Official Review · Reviewer_Hkfd · 2025-07-05

**Clarity:** 3
**Significance:** 3
**Originality:** 3
**Rating:** 2
**Confidence:** 5

**Summary:**

The paper introduces SIRIUS, a bootstrapped optimisation loop for multi-agent LLM systems. Starting from GPT-3.5-turbo or GPT-4o-mini, it instantiates domain roles (e.g., Physicist, Mathematician, Summariser), logs complete successful trajectories from datasets such as combined MMLU/GPQA/TheoremQA (Physics & Chemistry) and PubMedQA, rewrites failed ones with agent feedback, and fine-tunes each role on this “experience library.” Across reasoning and three negotiation games, SIRIUS lifts accuracy by 3 – 22 pp and improves bargaining outcomes versus single-agent, STaR, COMM, and TextGrad baselines while requiring no extra human labels.

**Questions:**

See Strengths And Weaknesses

**Ethical Concerns:**

["NO or VERY MINOR ethics concerns only"]

**Final Justification:**

The rebuttal satisfactorily adds open-source Llama-3 results (confirming gains are not GPT-specific) and discloses raw token counts, but it still omits a horizontal cost-effectiveness table that puts Single-Agent, STaR, COMM, TextGrad, and SIRIUS on the same $ scale, and it offers no experiment contrasting whole-trajectory supervision with explicit step-level credit in tasks requiring fine-grained rewards; because these two gaps leave the practical value and robustness of SIRIUS unclear, I maintain a reject (2/5) recommendation.

**Limitations:**

yes

**Quality:**

3

**Strengths And Weaknesses:**

The paper’s chief liabilities all stem from its infrastructure choices. First, every result hinges on proprietary GPT-3.5-turbo-0125 and GPT-4o-mini backbones; the authors themselves note that “the performance of SIRIUS is inherently tied to the capabilities of the underlying LLMs” and that any weaknesses in those models flow straight through the whole pipeline, leaving open-source substitutes untested and uncertain. Second, although SIRIUS learns from full successful trajectories to dodge hand-crafted rewards, it never resolves the classic multi-agent credit-assignment problem—the paper concedes that attributing success to individual actions “remains a complex issue” in unstructured language interactions. Finally, the system demands repeated multi-agent generation, feedback, rewriting, and round-robin fine-tuning; coupled with maintaining large per-role experience libraries and calling the OpenAI fine-tuning API each iteration, this makes the engineering and compute footprint non-trivial, yet the paper offers no cost-benefit analysis.

SIRIUS nonetheless delivers a neat, largely hands-off recipe for bootstrapping specialised agents: it stores whole successful dialogues in an experience library, automatically “cleans” failed trajectories via a feedback agent, and fine-tunes each role on this growing dataset, creating a self-improving loop that needs no manual reward shaping or step-level supervision. This role-specialisation plus joint optimisation yields consistent gains over strong baselines—up to 15.9% on College Chemistry and +17.8% on PubMedQA with GPT-3.5, and solid boosts in negotiation pay-offs and win-rates across three competitive games. The framework is task-agnostic, extending from science Q&A to adversarial bargaining, and its ablations show performance drops whenever a single SIRIUS agent is swapped out, underscoring genuine synergy among the roles

---

> ### Author Rebuttal · Authors · 2025-07-31
>
> We sincerely thank the reviewer for recognizing that SIRIUS offers an effective approach to bootstrapping and improving specialized agents via role-level fine-tuning. We have addressed your concerns below and hope these clarifications will assist you in re-evaluating and update the score:
>
> ### ` W1. Evaluation on Open source model`
>
> We thank the reviewer for highlighting this important point. We have conducted additional experiments on Llama-3.2-3B-instruct, which show promising improvements across all tasks.
>
> |   Model   |       Method       | College Physics | College Chemistry |   PubMedQA    |
> |:---------:|:------------------:|:----------------:|:------------------:|:-------------:|
> | **Llama-3.2-3B-instruct** | Single-Agent       | 26.79 ± 1.95     | 34.46 ± 2.28       | 54.07 ± 0.31  |
> |           | STaR                    | 27.41 ± 1.95   | 38.46 ± 1.49       | 54.93 ± 0.12  |
> |           | COMM               | 27.73 ± 1.43     | 37.44 ± 3.11       | 65.80 ± 0.87  |
> |           | TextGrad           | *Not supported* | – | – |
> |           | **SIRIUS**         | **29.60** ± 1.43 | **42.56** ± 2.28   | **68.27** ± 1.03 |
>
> We agree that demonstrating SIRIUS's effectiveness on open-source models is crucial for the community.  We appreciate the reviewer’s comment and hope this addresses the main concern.
>
> ### ` W2. Credit Assign Challenge`
>
> We appreciate the reviewer’s insightful comment. Our goal is not to solve the fundamental credit assignment problem directly, but to provide a practical and scalable approach for training multi-agent systems. By learning from complete successful trajectories, SIRIUS avoids the need for fine-grained rewards or step-level credit attribution, while partially addressing the challenge through targeted agent-level feedback during trajectory augmentation. We will revise the introduction and discussion to better clarify this point, and we aim to more directly explore the open challenge of credit assignment in multi-agent systems in future work.
>
> ### `W3. Cost analysis`
>
> Thank you for pointing this out. To support reproducibility and cost estimation, we report the approximate token usage at each major stage of the SIRIUS pipeline. Given that API prices for models like GPT‑4o‑mini, GPT‑3.5‑Turbo, and LLaMA‑3 are typically at or below $0.60 per million tokens, the overall cost is affordable. Considering the substantial performance gains (up to 21.88%), we believe the cost-performance trade-off is highly favorable and supports the practical viability of SIRIUS.
>
> | Domain  | Phase     | Category     | M tokens |
> |---------|-----------|--------------|---------|
> | College Physics      | generate  | solve        | 1.42    |
> |         |           | feedback     | 0.34    |
> |         |           | regenerate   | 15.98   |
> |         |           | finetune     | 1.12    |
> |         | eval      | eval         | 4.32    |
> |         |           | **total**    | **23.19** |
> | College Chemistry    | generate  |solve      | 0.80    |
> |         |           | feedback     | 0.16    |
> |         |           | regenerate   | 7.32    |
> |         |           | finetune     | 0.91    |
> |         | eval      | eval         | 2.41    |
> |         |           | **total**    | **11.60** |
> | PubMedQA  | generate  | solve          | 1.13    |
> |         |           | feedback     | 0.22    |
> |         |           | regenerate   | 7.11    |
> |         |           | finetune     | 0.88    |
> |         | eval      | eval         | 7.45    |
> |         |           | **total**    | **16.80** |
>
> This analysis will be included in the final version for completeness and clarity.
>
> ## **Summary**
>
> We hope our answers address your concerns on (1) open-source evaluation, (2) credit assignment, and (3) cost analysis.  We also sincerely thank the reviewer for recognizing our novel and effective framework, the strength of our empirical evaluation, and its task-agnostic design. Finally, we truly appreciate your reconsideration of our work in light of our responses. Thank you once again for your valuable insights!

---

> > ### Comment · Reviewer_Hkfd · 2025-08-06
> >
> > Thanks for the thoughtful rebuttal and the new results. To make the paper even more convincing, I would still recommend adding a comprehensive cost–benefit table: report, for Single-Agent, STaR, COMM, TextGrad, and SIRIUS, the total token usage and corresponding dollar cost (specifying the backbone model and its unit price), and compute a unified metric such as “cost per 1% accuracy gain” or “cost per 1-point negotiation utility gain.” Only with this horizontal comparison can readers fairly judge the true value of SIRIUS.
> >
> > In addition, because whole-trajectory supervision may break down when tasks demand fine-grained step-level rewards or penalties (e.g., real-time control or multi-round revenue sharing), it would be helpful to include at least one synthetic environment where explicit step-level rewards are available and to compare “pure trajectory supervision” against “trajectory + explicit credit-assignment.” This would clarify the boundary conditions of the current framework and highlight a concrete path for future work.
> >
> > Base on the current rebuttal, I still insist on rejecting this paper if these key problems haven't been fully addressed. While I am open to modify the rating if any further insightful response.

---

> > > ### Author Response · Authors · 2025-08-07
> > > **Explicit step-level rewards**
> > >
> > > Credit assignment is indeed a challenging and largely unsolved problem. As we mentioned in introduction, an approach in traditional reinforcement learning is to use a counterfactual baseline to estimate the expected return for each state–action pair. However, this method requires sampling a large number of trajectories at every step, and the computational cost grows exponentially with the number of agents. This is often prohibitive, especially given that the assigned credit is merely an estimate. A similar line of work involves learning *process rewards*, which requires extensive human-labeled data and the joint training of a reward model to supervise intermediate steps. This approach also incurs substantial annotation and training costs.
> > >
> > > Instead of relying on explicit step-level supervision, our method solve this challenge in a more practical way:
> > >
> > > - For questions where the final outcome is correct, we naturally assign equal credit to all agents, as each contributes to the successful trajectory.
> > > - For questions with incorrect outcomes, we *generate feedback with* ground truth for each agent start from agent 1. We then regenerate the trajectory based on the feedback and reassess credit assignment according to the final reward.
> > >
> > > This approach avoids the inefficiency of massive trajectory sampling to deal with credit assignment problem and enables a form of retrospective credit attribution that is both scalable and effective in identifying the agents responsible for failure.
> > >
> > > **Moreover, our second Actor-Critic setting can be seen as an attempt at explicit  step-level credit assignment**. In this setup, the Actor Agent first generates a solution, which is then evaluated by the Judgment Agent for correctness. If the solution is deemed incorrect, the Critic Agent analyzes the original response and provides feedback—without access to the ground truth. In this system, credit is naturally attributed at each step:
> > >
> > > - the Actor is credited based on whether it provides a correct initial solution,
> > > - the Judgment Agent based on whether it correctly determine initial solution’s correctness.
> > > - and the Critic Agent based on whether it provides useful feedback for Actors.
> > >
> > > **These step-level credits are well-defined at trajectory evaluation time and do not rely solely on the system's final answer correctness.** Our ablation study further underscores the contribution of each individual agent in the overall performance of SIRIUS.
> > >
> > > We will include this comparison and analysis to clarify the boundary of SIRIUS’s applicability and highlight concrete future directions.
> > >
> > > ### **Summary**
> > > We hope our responses address the reviewer’s concerns and we will definitely incorporate your valuable suggestions in our revised version.We believe these additions will significantly enhance the rigor and impact of our work.
> > >
> > > We sincerely appreciate your willingness to reconsider the rating and thank you once again for your constructive and thoughtful feedback.

---

> ### Author Response · Authors · 2025-08-07
> **Cost–Benefit Analysis**
>
> Thank you again for your thoughtful feedback and helpful suggestions. We appreciate the opportunity to further clarify and strengthen our work.
>
> ### `1. Cost–Benefit Analysis`
> In response to your suggestion, we have calculated the total token usage and corresponding dollar cost for all compared baselines based on our existing experimental logs. The costs are estimated using public API pricing and reported in the following table, alongside performance metrics and unified metrics: "improvement per extra dollar" and “extra cost per 1% accuracy gain.”
>
> ```
> Improvement per extra dollar = improvement / (SIRIUS's cost- baseline's cost)
>
> Extra cost per 1% accuracy gain = (SIRIUS's cost- baseline's cost) / improvement
> ```
>
> ### Model: Llama-3.2-3B-instruct
>
> | Dataset | Settings| Cost | Score | Improvement | Improvement per Dollar | Cost per 1% Acc Gain |
> |-----------------|:-------------:|:----:|:-----:|:-----------:|:-----------------------:|:--------------------------:|
> | College-Physics | Single| 0.06 | 26.79 | 10.49%| 7.28% | 0.14 |
> | | STaR| 0.22 | 27.41 | 40.31%| 31.50%| 0.03 |
> | | COMM| 0.26 | 27.73 | 38.69%| 31.21%| 0.03 |
> | | TextGrad|Not supported|
> | | SIRIUS| 1.50 | 29.60 | / | /| /|
> | College-Chemistry | Single| 0.03 | 34.36 | 23.86%| 30.17%| 0.03 |
> | | STaR| 0.12 | 38.46 | 42.82%| 61.18%| 0.02 |
> | | COMM| 0.15 | 37.44 | 46.71%| 69.72%| 0.01 |
> | | TextGrad|Not supported|
> | | SIRIUS| 0.82 | 42.56 | / | /| /|
> | PubMed| Single| 0.06 | 54.07 | 26.26%| 21.40%| 0.05 |
> | | STaR| 0.32 | 54.93 | 24.29%| 25.04%| 0.04 |
> | | COMM| 0.45 | 65.80 | 3.75% | 4.47% | 0.22 |
> | | TextGrad| Not supported|
> | | SIRIUS| 1.29 | 68.27 | / | /| /|
>
>
> ### Model: gpt-3.5-turbo
> | Dataset | Settings| Cost | Score | Improvement | Improvement per Dollar | Cost per 1% Acc Gain |
> |-----------------|:-------------:|:----:|:-----:|:-----------:|:-----------------------:|:--------------------------:|
> |**College-Physics**|Single|0.30|25.55|32.93%|0.25%|0.25|
> ||STaR|3.47|30.84|10.10%|2.02%|0.49|
> ||COMM|3.54|28.97|17.20%|3.49%|0.29|
> ||TextGrad|4.19|32.09|5.83%|1.36%|0.73|
> ||SIRIUS|8.46|33.96|/|/|/|
> |**College-Chemistry**|Single|0.24|40.00|39.74%|5.30%|0.19|
> ||STaR|3.05|45.64|22.47%|4.79%|0.21|
> ||COMM|2.13|47.69|17.20%|3.07%|0.33|
> ||TextGrad|5.28|41.54|34.57%|14.05%|0.07|
> ||SIRIUS|7.74|55.90|/|/|/|
> |**PubMed**|Single|4.20|57.53|30.94%|1.89%|0.53|
> ||STaR|12.89|64.20|17.34%|2.27%|0.44|
> ||COMM|7.65|72.27|4.24%|0.33%|3.04|
> ||TextGrad|failed |to | parse |instructions|
> ||SIRIUS|20.53|75.33|/|/|/|
>
> ### Model: gpt-4o-mini
> |Dataset|Settings|Cost|Score|Improvement|ImprovementperDollar|Cost per 1% Acc Gain|
> |------------------|------------------|:-------:|:--------:|:-------------:|:---------------------------:|:-------------------------------:|
> |**College-Physics**|Single|0.27|38.63|24.79%|4.73%|0.21|
> ||STaR|2.36|42.99|12.13%|3.86%|0.26|
> ||COMM|1.59|42.37|13.78%|3.52%|0.28|
> ||TextGrad|2.47|39.25|22.81%|7.52%|0.13|
> ||SIRIUS|5.51|47.35|/|/|/|
> |**College-Chemistry**|Single|0.14|40.00|64.83%|21.49%|0.05|
> ||STaR|1.11|48.21|36.78%|17.94%|0.06|
> ||COMM|0.84|49.23|33.93%|14.64%|0.07|
> ||TextGrad|1.21|49.74|32.55%|16.68%|0.06|
> ||SIRIUS|3.16|56.41|/|/|/|
> |**PubMed**|Single|1.82|63.87|15.34%|2.07%|0.48|
> ||STaR|4.93|65.93|11.73%|2.72%|0.37|
> ||COMM|3.06|70.27|4.84%|0.78%|1.28|
> ||TextGrad|4.81|66.20|11.28%|2.55%|0.39|
> ||SIRIUS|9.24|73.67|/|/|/|
>
> SIRIUS yields an average improvement of 33.13%, 3.35%, and 8.15% per additional dollar spent on LLaMA 3.2 3B, GPT-3.5, and GPT-4o-mini, respectively. We believe this comparison makes the cost–benefit trade-offs of each method more transparent and highlights the practical efficiency of SIRIUS in balancing performance and resource usage.

---

### Note · Authors · 2025-08-13

Dear ACs and Reviewers,

We sincerely thank you for the time and effort in evaluating our paper. To facilitate the next stage of discussion, we summarize the key points from the rebuttal as below:

### **Key Strengths Highlighted by Reviewers**

- `Reviewer Hkfd`: **novel and effective framework**, **comprehensive evaluation**, and **task-agnostic design** of SIRIUS, providing an effective approach for improving specialized agents.
- `Reviewer AYyk`: **strong and effective approach**, **comprehensive evaluation** against strong baselines, **insightful ablation findings**, **novel evidence** and **comprehensive study** across multiple architectures.
- `Reviewer HHDU`: focus on a **crucial and novel problem**, novel use of negative samples to improve **data utilization** and **training efficiency**.
- `Reviewer Ko9b`: focus on a **challenging and relevant** problem, **diverse and well-chosen settings**, **robust generalization** results, and **comprehensive ablations** that rigorously validate the key design choices

### **How We Addressed Reviewer Concerns**

`Reviewer Hkfd`: **(1)** We provided additional open-source evaluation on Llama-3.2-3B-instruct. **(2)** Reported Cost–benefit analysis for all baselines, and the unified metric “extra cost per 1% accuracy gain” requested by the reviewer, highlighting SIRIUS’s cost-efficiency. **(3)** Addressed the credit assignment challenge by explaining SIRIUS’s efficient alternative compared to costly traditional methods. In response to the reviewer’s request for explicit step-level rewards, we clarified our Actor–Critic setting already provides such step-level credit without relying solely on final-answer correctness, thereby clarifying the framework’s scope and pointing to concrete future directions.

`Reviewer AYyk`: We added the MIPROv2 method in DSPy as an additional baseline, clarified the intended usage and limitations of BootstrapFT, and noted that Table 4 already contains the requested setting.

For `Reviewers HHDU` and `Reviewers ko9b`,we provided detailed responses thoroughly addressed their concerns. Both reviewers acknowledged these revisions and agreed the paper is suitable for acceptance, noting its positive impact on the field.

We will definitely incorporate all additional evaluations, analyses, and your valuable suggestions into the final version. Once again, we sincerely thank the Area Chairs and Reviewers for their time and constructive feedback.

With sincere respect and gratitude,

Authors

---

### Decision · Program_Chairs · 2025-09-17

**Decision:**

Accept (poster)

**Comment:**

The paper proposes SiriuS, a self-improving framework for multi-agent LLM systems. It builds an experience library by retaining successful reasoning trajectories and augmenting failed ones with corrective feedback, then fine-tunes specialized agents on this evolving dataset. SiriuS is evaluated across scientific QA (MMLU, GPQA, TheoremQA, PubMedQA) and negotiation games, showing consistent improvements of 2.9%–21.9% over strong baselines such as Single-Agent, STaR, COMM, and TextGrad.

### Strengths of the paper:

- The reviewers (AYyk, HHDU, ko9b) highlight the novelty and practicality of the trajectory-based self-improvement loop, noting its broad applicability across collaborative problem solving, actor–critic feedback, and competitive multi-agent negotiation.

- The work is technically sound and well-structured, with ablations demonstrating the necessity of each design choice (role specialization, trajectory augmentation, multi-agent fine-tuning).

- The reviewers appreciate the effective use of failed trajectories to enrich training data, which improves data efficiency and stability.

- The empirical results are strong, consistently outperforming baselines across multiple tasks and model backbones, with meaningful improvements in reasoning and negotiation performance.

- The paper is generally well-written, easy to follow, and offers clear insights into both opportunities and limitations of multi-agent optimization.

### Weaknesses of the paper:

- The reviewer Hkfd emphasizes the lack of a fundamental solution to the credit assignment problem: SiriuS sidesteps it rather than resolving attribution of success to individual actions.

- Reviewers note the dependency on ground-truth signals for trajectory augmentation, limiting applicability to domains with labeled supervision.

- Cost and scalability concerns remain: the iterative process of multi-agent generation, regeneration, and fine-tuning requires substantial compute, and while cost–benefit analyses were later provided, questions of long-term scalability persist.

- Evaluation scope is somewhat narrow, omitting more challenging or practical agent settings (e.g., deep research tasks, real-world planning). Some baselines (e.g., BootstrapFT in DSPy) required clarification.

- Presentation could be more precise on methodological details (e.g., success thresholds, oracle reliance in augmentation).

### Primary reasons for Accept (Poster)

The primary reasons for recommending Accept (Poster) are that SiriuS introduces a general and practical bootstrapped learning loop for multi-agent systems, demonstrating clear novelty in systematically transforming both successful and failed trajectories into reusable training data. The framework is well-motivated, empirically validated across diverse tasks, and provides consistent gains over strong baselines. While the reliance on ground-truth signals and unresolved credit assignment limit broader applicability, the rebuttal addressed major reviewer concerns by adding open-source evaluations (LLaMA-3.2), cost–benefit comparisons, and clarifying Actor–Critic feedback. Overall, the paper makes a timely and meaningful contribution to multi-agent LLM optimization, justifying acceptance as a poster.

### Summary of the discussion and rebuttal

The authors provided detailed and constructive responses. For R-Hkfd, they added open-source results (LLaMA-3.2), disclosed token-level cost analyses, and clarified retrospective credit attribution strategies and Actor–Critic supervision, partially addressing concerns about cost and credit assignment. For R-AYyk, they compared SiriuS against DSPy’s BootstrapFT, showing consistent improvements, and clarified the distinction between pipeline optimization and interactive agent training. For R-HHDU, they reported augmentation coverage (32%–74%) across datasets and demonstrated improvements with newer models like GPT-4o-mini, leading to an upgraded score. For R-ko9b, they clarified the Actor–Critic setup and acknowledged the limitation of oracle-assisted augmentation, while highlighting broader applicability in negotiation settings. Overall, the rebuttal strengthened empirical support and methodological clarity, and several reviewers raised their scores, leading to consensus on poster acceptance.